# Spectral-inspired Operator Learning with Limited Data and Unknown Physics

## Abstract

Learning PDE dynamics from limited data with unknown physics is challenging. Existing neural PDE solvers either require large datasets or rely on known physics (e.g., PDE residuals or handcrafted stencils), leading to limited applicability. To address these challenges, we propose **S**pectral-**I**nspired **N**eural **O**perator (SINO), which can model complex systems from just 2-5 trajectories, without requiring explicit PDE terms. Specially, SINO automatically captures both local and global spatial derivatives from frequency indices, enabling a compact representation of the underlying differential operators in physics-agnostic regimes. To model non-linear effects, it employs a $\Pi$-block that performs multiplicative operations on spectral features, complemented by a low-pass filter to suppress aliasing. Extensive experiments on both 2D and 3D PDE benchmarks demonstrate that SINO achieves state-of-the-art performance, with improvements of 1–2 orders of magnitude in accuracy. Particularly, with only 5 training trajectories, SINO outperforms data-driven methods trained on 1000 trajectories and remains predictive on challenging out-of-distribution cases where other methods fail.

## 1 Introduction

Partial differential equations (PDEs) govern the evolution of numerous physical systems, from heat transfer (Pletcher et al., 2012) and chemical reactions (Ghergu & Radulescu, 2011) to fluid dynamics (Holton & Hakim, 2013). Accurately modeling and simulating such PDE-governed systems is essential for the understanding of natural phenomena and engineering applications. However, in many scientific computing scenarios, we face two fundamental challenges: **(i) only a few trajectories of data are available**, and **(ii) the governing physics is unknown**. While traditional methods offer strong stability and interpretability, their reliance on a deep understanding of the physical system limits their application in scenarios where the underlying physics is unknown.

In recent years, with the development of deep learning, data-driven neural PDE solvers have been proposed to address the limitations of traditional numerical methods (Azizzadenesheli et al., 2024). Representative works include DeepONet (Lu et al., 2021), FNO (Li et al., 2021), and DPOT (Hao et al., 2024). By leveraging the expressive power of neural networks and large datasets, these methods learn mappings between function spaces. As a result, they can reduce computational costs under coarse spatiotemporal resolutions and do not require explicit knowledge of the underlying PDEs. While offering notable advantages, such approaches remain highly data-hungry. In many scientific and engineering domains, obtaining sufficient training data requires expensive physical experiments (e.g., wind tunnel testing or combustion experiments), which severely limits their practical applications (Parente & Swaminathan, 2024; Li et al., 2024a; Zhang et al., 2025b).

To reduce the data demands of data-driven solvers, physics-aware methods incorporate physical knowledge into deep learning models, yielding two main paradigms: physics-informed and physics-encoded (Faroughi et al., 2022). Physics-informed methods, such as PINNs (Raissi et al., 2019) and PINO (Li et al., 2024c), add soft constraints by including PDE residuals in the loss function. They require explicit forms of the governing PDEs and often suffer from training instability (Krishnapriyan et al., 2021). On the other hand, physics-encoded methods, such as PeRCNN (Rao et al., 2023; 2022), P$^2$C$^2$Net (Wang et al., 2024c), and TSM (Sun et al., 2023), embed physical rules directly into the network architecture, e.g., hardcoding finite difference stencils as convolutional kernels (Rao et al., 2023). When designing these models, one still needs complete or partial PDE terms

as prior knowledge. Moreover, since many architectures borrow from local numerical schemes such as the finite difference method (FDM) or the finite volume method (FVM), they inherit limitations in handling high-order derivatives and global interactions (McGreivy & Hakim, 2024).

To address these challenges, we propose **S**pectral-**I**nspired **N**eural **O**perator (SINO), which can accurately model complex systems from as few as 2-5 trajectories without any explicit PDE terms. The design of SINO is inspired by spectral methods, which are well known for their ability to capture global information with exponential accuracy (Jingrun et al., 2022). Specifically, we propose a Frequency-to-Vector (Freq2Vec) module that maps each frequency index into a learnable embedding, mimicking the derivative multipliers in spectral methods without relying on predefined operators. To further capture nonlinear dynamics, we employ a $\Pi$-block that models multiplicative interactions, together with a low-pass filter to mitigate aliasing effects. By inheriting the structure of spectral methods, SINO achieves stronger generalization from limited observations and demonstrates robust out-of-distribution (OOD) performance compared to data-driven approaches, while also handling global interactions and high-order derivatives more effectively than other physics-aware methods. **In summary, we make the following contributions**:

**(1) Novel architecture**. We propose SINO, a data-efficient neural operator for modeling complex spatiotemporal dynamics without requiring prior knowledge of PDE terms. We propose two key modules: Freq2Vec simulates frequency domain derivative multipliers, and the nonlinear operator block captures complex interactions through efficient multiplicative operations.

**(2) State-of-the-art (SOTA) performance**. SINO achieves SOTA performance across multiple 2D and 3D PDE benchmarks, delivering improvements of 1–2 orders of magnitude in accuracy compared to baselines. To our knowledge, SINO is among the first physics-aware methods capable of simulating globally coupled systems such as the NSEs without requiring explicit PDE terms.

**(3) Data efficiency and OOD handling**. Due to the spectral-inspired design, SINO performs genuine operator learning rather than data memorization. With only 5 training trajectories, SINO outperforms data-driven methods trained on 1000 trajectories and remains predictive in out-of-distribution (OOD) tests where other methods fail.

## 2 RELATED WORK

**Data-driven neural PDE solvers.** When provided with abundant data, data-driven methods learn mappings between function spaces, bypassing the need for explicit PDE formulation. Representative examples include the FNO (Li et al., 2021) and its variants (Tran et al., 2023; Wen et al., 2022), DeepONet (Lu et al., 2021) and its variants (Venturi & Casey, 2023; Lee et al., 2023). Transformer-based models (Wu et al., 2024; Li et al., 2024b) and graph-based architectures (Pfaff et al., 2021; Brandstetter et al., 2022) extend neural PDE solvers to handle complex geometries by operating on mesh or graph representations. Recently, PDE foundation models aim to solve multiple PDE families within a single framework, achieving broad applicability across different physical systems (Hao et al., 2024; Zhang et al., 2024; Hang et al., 2024). Thanks to their rapid inference ability, these data-driven solvers have found broad scientific computing applications, such as weather forecasting (Zhang et al., 2023), control systems (Hu et al., 2025), and geometric design (Wang et al., 2024b). However, a major drawback of data-driven PDE solvers is their heavy reliance on large datasets, which in many applications are costly to obtain through expensive physical experiments (Parente & Swaminathan, 2024; Li et al., 2024a). In data-scarce regimes, their generalization performance degrades substantially. In contrast, SINO uses a spectral-inspired framework to embed physical structure, enabling effective training from limited data and robust OOD generalization.

**Physics-aware neural PDE solvers.** Unlike data-driven methods, physics-aware methods aim to reduce data dependency by embedding physical priors into the learning process or architecture design (Faroughi et al., 2022). Based on how they incorporate physical laws, physics-aware methods can be categorized into physics-informed and physics-encoded methods. Physics-informed methods integrate PDE residuals (Wang et al., 2021) or numerical schemes (e.g., FDM (Huang et al., 2023), spectral methods (Du et al., 2024), particle methods (Zhang et al., 2025a)) into the loss function, but require complete knowledge of the governing PDEs and often suffer from training instability (Krishnapriyan et al., 2021). Physics-encoded methods incorporate physical structure in a hard-constrained manner, such as using FDM (Long et al., 2019; Rao et al., 2023) or FVM (Kochkov

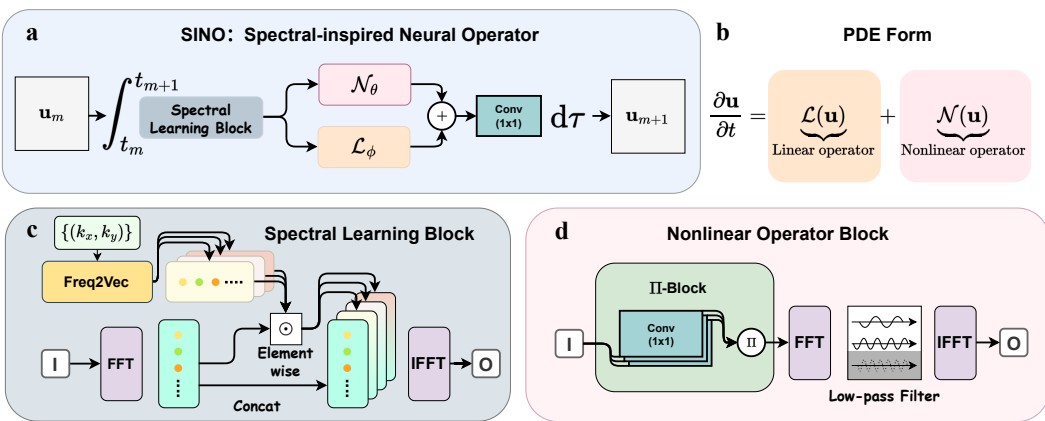

Figure 1: **The framework of SINO.** (a) The overall architecture. (b) The PDE formulation. (c) The structure of the Spectral Learning Block (SLB). (d) The structure of the Nonlinear Operator Block (Π-Block + low-pass filter).

et al., 2021; Sun et al., 2023), yet depend on specific PDE terms and are constrained by their local receptive field. Additionally, recent physics-encoded approaches that incorporate spectral methods have shown promise in improving global modeling but still require full or partial knowledge of the governing PDE terms, especially when tackling complex systems such as the NSEs (Dresdner et al., 2023; Li et al., 2025). Unlike these hybrid approaches, SINO not only captures global information in the frequency domain but also requires no prior knowledge of specific PDE terms.

**Physics-aware neural PDE solvers.** Unlike data-driven methods, physics-aware methods aim to reduce data dependency by embedding physical priors into the learning process or architecture design (Faroughi et al., 2022). Based on how they incorporate physical laws, physics-aware methods can be categorized into physics-informed and physics-encoded methods. Physics-informed methods integrate PDE residuals (Wang et al., 2021) or numerical schemes (e.g., FDM (Huang et al., 2023), spectral methods (Du et al., 2024), particle methods (Zhang et al., 2025a)) into the loss function, but require complete knowledge of the governing PDEs and often suffer from training instability (Krishnapriyan et al., 2021). Physics-encoded methods incorporate physical structure in a hard-constrained manner, such as using FDM (Long et al., 2019; Rao et al., 2023) or FVM (Kochkov et al., 2021; Sun et al., 2023), yet depend on specific PDE terms and are constrained by their local receptive field. Additionally, recent physics-encoded approaches that incorporate spectral methods have shown promise in improving global modeling but still require full or partial knowledge of the governing PDE terms, especially when tackling complex systems such as the NSEs (Dresdner et al., 2023; Li et al., 2025). Unlike these hybrid approaches, SINO not only captures global information in the frequency domain but also requires no prior knowledge of specific PDE terms.

## 3 METHODOLOGY

### 3.1 PROBLEM SETTING AND PRELIMINARY

This paper addresses the problem of modeling PDE-governed systems from limited data when PDE terms are unknown. Consider a system defined on $\Omega \subset \mathbb{R}^d$, whose state $\mathbf{u}(\mathbf{x}, t; \boldsymbol{\mu}) \in \mathbb{R}^C$ satisfies

$$\partial_t \mathbf{u}(\mathbf{x}, t; \boldsymbol{\mu}) = \mathcal{L}(\mathbf{u}(\mathbf{x}, t; \boldsymbol{\mu})) + \mathcal{N}(\mathbf{u}(\mathbf{x}, t; \boldsymbol{\mu})). \quad (1)$$

Here, $\mathbf{u}$ denotes the state field, $\boldsymbol{\mu}$ represents PDE parameters, and $\mathcal{L}(\cdot)$ and $\mathcal{N}(\cdot)$ are *unknown* linear and nonlinear differential operators, which include multiple spatial derivatives of various orders. When the PDE formulation in Eq. 1 is known, spectral methods are among the most accurate and fastest numerical methods on regular domains because they exhibit exponential convergence and capture global information (Shen et al., 2011; McGreivy & Hakim, 2024). Given the state $\mathbf{u}_m(\mathbf{x})$ at time $t_m$, a spectral solver advances to $t_{m+1}$ in three steps.

*Firstly,* the state is transformed to frequency space, and all spatial derivatives are computed. In detail, the Fourier transform $\mathcal{F}$ converts $\mathbf{u}_m$ into spectral coefficients $\widehat{\mathbf{u}}_m$ as follows:

$$\widehat{\mathbf{u}_m}(\mathbf{k}) = \mathcal{F}\big[\mathbf{u}_m(\mathbf{x})\big] = \int_\Omega \mathbf{u}_m(\mathbf{x})\, e^{-\,\mathrm{i}\,\mathbf{k}\cdot\mathbf{x}}\,\mathrm{d}\mathbf{x}, \tag{2}$$

where $\mathbf{k}$ is the frequency indexing corresponding to different frequency components. In the frequency domain, each derivative operator can become an exact multiplier. For example, the Laplacian operator $\nabla^2 \mathbf{u}_m$ satisfies $\widehat{\nabla^2 \mathbf{u}_m}(\mathbf{k}) = -\|\mathbf{k}\|_2^2\,\widehat{\mathbf{u}_m}(\mathbf{k})$ in the frequency domain.

*Secondly,* each derivative is transformed back to the spatial domain via the inverse Fourier transform $\mathcal{F}^{-1}$ and combined to form the linear and nonlinear terms of the PDE. For linear terms, the corresponding spectral derivatives can be directly scaled by their coefficients and then summed. For example, the diffusion term $\nu \nabla^2 \mathbf{u}_m$ can be denoted as $\nu \mathcal{F}^{-1}[\widehat{\nabla^2 \mathbf{u}_m}]$. For nonlinear terms, the computation requires calculating element-wise products in the spatial domain. For instance, the convection term $\nabla \mathbf{u}_m \cdot \mathbf{u}_m$ can be calculated as $\mathcal{F}^{-1}[\widehat{\nabla \mathbf{u}_m}] \cdot \mathcal{F}^{-1}[\widehat{\mathbf{u}_m}]$.

*Thirdly,* a high-order time-stepping scheme (e.g., a fourth-order Runge-Kutta (RK4) scheme) is deployed to obtain $\mathbf{u}_{m+1}(\mathbf{x})$.

## 3.2 Overall architecture

Motivated by spectral methods, we propose SINO to model nonlinear systems from limited data when all PDE terms are unknown. Similar to a spectral solver, SINO follows three steps (Fig. 1a).

*Firstly,* SINO maps the input state $\mathbf{u}_m$ to the frequency domain via the Fourier transform and uses a Spectral Learning Block (SLB) to simulate various spatial derivatives (Fig. 1c). Because the exact PDE terms are unknown, we employ a Frequency-to-Vector (Freq2Vec) module that learns how to represent each derivative from its frequency indexing $\mathbf{k}$ under the supervision of limited data.

*Secondly,* the learned representations are transformed through linear ($\mathcal{L}_\phi$) and nonlinear ($\mathcal{N}_\theta$) parts and then recombined by a $1 \times 1$ convolution, where $\phi$ and $\theta$ denote the parameters of SINO. Specifically, in the nonlinear branch, we use a $\Pi$-block to model the nonlinear interactions via element-wise products and a low-pass filter to prevent aliasing (Fig. 1d).

*Thirdly,* the combined right-hand side is advanced in time using an RK4 scheme to obtain $\mathbf{u}_{m+1}$. As a result, the overall framework of SINO can be described as

$$\mathbf{u}_{m+1} = \int_{t_m}^{t_{m+1}} \mathrm{Conv}_{1x1}\left(\mathcal{L}_\phi(\mathrm{SLB}(\mathbf{u}_\tau)) + \mathcal{N}_\theta(\mathrm{SLB}(\mathbf{u}_\tau))\right)\mathrm{d}\tau. \tag{3}$$

## 3.3 Architecture components of SINO

We herein detail the network architecture of SINO, including the spectral learning block, the linear block, and the nonlinear block.

**Spectral learning block.** In spectral methods, spatial derivatives are computed exactly in the frequency domain, such as $\widehat{\partial_x^n \mathbf{u}}(\mathbf{k}) = (\mathrm{i}k_x)^n \cdot \widehat{\mathbf{u}}(k)$. When PDE terms are unknown, we design a spectral learning block to learn various spatial derivatives in the frequency domain. In detail, we propose a key module named Frequency-to-Vector (Freq2Vec). This module learns a latent representation $\psi(\mathbf{k})$ for each frequency indexing $\mathbf{k}$ via a shared MLP, i.e., $\psi(\mathbf{k}) = \mathrm{MLP}(\mathbf{k})$. The output $\psi(\mathbf{k})$ encodes the operator action at indexing $\mathbf{k}$ and plays a role analogous to the exact multipliers (e.g., $(\mathrm{i}k_x)^n$) used in classical spectral methods. To approximate the unknown derivative operator $\mathcal{D}$, SINO computes a element-wise product in the frequency domain, i.e., $\widehat{\mathcal{D}\mathbf{u}}(\mathbf{k}) = \psi(\mathbf{k}) \cdot \widehat{\mathbf{u}}(\mathbf{k})$, which mimics the way spectral methods apply multipliers to represent differential operators. The result is then transformed back to the spatial domain: $\mathcal{D}\mathbf{u}(\mathbf{x}) = \mathcal{F}^{-1}[\widehat{\mathcal{D}\mathbf{u}}(\mathbf{k})]$.

A key advantage of this spectral formulation is that SINO not only learns standard spatial differential operators with high accuracy, but also naturally captures globally coupled relations. For instance, the velocity-vorticity formulation in 2D NSE can be expressed in a simple closed form. Given vorticity

$\omega = \partial_x \mathbf{u}_y - \partial_y \mathbf{u}_x$, the velocity can be recovered via the Biot–Savart law (Saffman, 1995):

$$\widehat{\mathbf{u}}(\mathbf{k}) = \frac{i\,\mathbf{k}^\perp}{\|\mathbf{k}\|_2^2}\,\widehat{\omega}(\mathbf{k}), \quad \mathbf{k}^\perp := (-k_y, k_x), \tag{4}$$

which corresponds to a simple spectral multiplier. Such relations can in principle be approximated by Freq2Vec due to its expressive capacity. By contrast, other physics-encoded methods (Wang et al., 2024c; Yan et al., 2025) built on local discretization schemes must solve a Poisson equation in Eq. 4, thereby requiring detailed PDE terms.

**Linear operator block.** After computing the spectral derivatives via the spectral learning block, we use a linear operator $\mathcal{L}_\phi$ to combine them with simple linear channel mixing. Concretely, the set of derived features is passed through a $1 \times 1$ convolution that learns to weight and sum these channels.

**Nonlinear operator block.** Unlike the linear block's simple channel mixing, the nonlinear operator block uses multiplicative interactions to emulate nonlinear PDE terms (Fig. 1d). Given the calculated spatial derivative features $\mathcal{D}\mathbf{u}(\mathbf{x})$, we design the $\Pi$-block as follows. We first map the derivative features into $P$ channels via a $1 \times 1$ convolution and then perform element-wise products across these channels to model nonlinear interactions:

$$\mathbf{v}_\Pi(\mathbf{x}) = \prod_{p=1}^{P} (W_p\,\mathcal{D}\mathbf{u}(\mathbf{x}) + b_p), \tag{5}$$

where $W_p$ and $b_p$ denote the weights and biases. This $\Pi$-block offers a lightweight yet powerful mechanism for modeling nonlinear interactions through element-wise products, which can naturally express various nonlinear terms (e.g., the convection term $\mathbf{u} \cdot \nabla \mathbf{u}$). As shown later in Fig. 4, feature maps from the $\Pi$-block closely align with ground-truth differential operators of the underlying PDEs, suggesting that SINO implicitly learns physically meaningful operators.

However, nonlinear interactions can create frequencies above the Nyquist limit, causing aliasing, where high frequencies fold into low ones and introduce spurious noise (Kravchenko & Moin, 1997). To mitigate this effect, we perform de-aliasing in the frequency domain using the classical low-pass filter with a $2/3$-rule. Specifically, we apply a Fourier transform to the nonlinear output and zero out high-frequency modes exceeding the $2/3$ cutoff:

$$\widetilde{\mathbf{v}}_\Pi(\mathbf{k}) = \mathbf{M}(\mathbf{k}) \cdot \mathcal{F}[\mathbf{v}_\Pi(\mathbf{x})](\mathbf{k}), \quad \mathbf{M}(\mathbf{k}) = \begin{cases} 1, & \|\mathbf{k}\|_\infty \leq 2/3k_{\max}, \\ 0, & \text{otherwise,} \end{cases} \tag{6}$$

where $k_{\max}$ denotes the highest frequency mode. This ensures that the nonlinear output does not distort the solution with aliased frequencies. Finally, we map the de-aliased representation back to the spatial domain. By combining $\Pi$-block with the low-pass filter, the nonlinear block in SINO captures complex operator interactions efficiently and enjoys strong numerical stability.

### 3.4 RELATION TO OTHER NEURAL SPECTRAL METHODS

SINO differs from existing spectral methods in both *what* it learns in the frequency domain and *how* it introduces nonlinearity. In the frequency domain, FNO (Li et al., 2021) uses a large spectral kernel to directly mix coefficients; SNO (Fanaskov & Oseledets, 2023), AFNO (Guibas et al., 2022), and DPOT (Hao et al., 2024) apply MLPs directly to frequency-domain feature maps, transforming the learned coefficients with a black-box mapping; LSM projects the dynamics into a compact latent space and performs its basis operations. While expressive, these designs largely behave as black boxes in the spectral domain and offer limited physical meaning, which makes them struggle in few-shot settings. In contrast, SINO learns embeddings over frequency indices $k$ with Freq2Vec, encoding the prior that derivative multipliers are functions of $k$. Coupled with a simple multiplicative $\Pi$-block, SINO's intermediate features align well with the ground-truth physical operators such as $\partial_x \omega$ and $\Delta \omega$ (Fig. 4), thereby yielding strong inductive biases for learning the underlying physics.

## 4 THEORETICAL INSIGHTS OF SINO

We now provide theoretical insights into why SINO performs well with limited data, focusing on its approximation ability and generalization behavior. In essence, SINO is a **compact yet expressive** neural operator that balances approximation accuracy and generalization.

For a physical field $\mathbf{u}$, one step of SINO corresponds to predicting the right-hand side (RHS) of the PDE, which involves linear and nonlinear combinations of spatial derivatives of $\mathbf{u}$. The following theorem reveals that SINO can approximate a broad class of such operators (proof in Appendix B):

**Theorem 1 (Approximation ability of SINO)** *Let $\Omega = [0,1)^d$ denote the $d$-torus and define the set $\mathcal{U} := \{\mathbf{v} : \Omega \to \mathbb{R}^C$ periodic, bandlimited to $\|\mathbf{k}\|_\infty \le k_{\max}, \|\mathbf{v}\|_{L^2} \le B\}$ for some $B > 0$. Let $\mathbf{u} : \Omega \to \mathbb{R}^C$ be any function with $\mathbf{u} \in \mathcal{U}$. Suppose the RHS of the PDE can be expressed as*

$$\partial_t \mathbf{u} = \mathcal{R}(\mathbf{u}) := \sum_{j=1}^{J} \prod_{i=1}^{n_j} \mathcal{D}_{j,i} \mathbf{u},$$

*where each $\mathcal{D}_{j,i}$ is a linear differential operator of finite order. Then for any $\epsilon > 0$, there exists a SINO model $\mathcal{S}_\theta$ such that*

$$\sup_{\mathbf{u} \in \mathcal{U}} \big\| \mathcal{S}_\theta(\mathbf{u}) - \mathcal{R}(\mathbf{u}) \big\|_{L^2} < \epsilon.$$

Theorem 1 shows that SINO can approximate nonlinear PDE operators constructed from linear derivatives and finite multiplicative interactions, covering a wide range of reaction-advection-diffusion systems. Beyond approximation, classical learning theory (Mohri et al., 2018) implies that the generalization gap scales as $\mathcal{O}(\sqrt{C(\mathcal{H})/N})$, where $C(\mathcal{H})$ denotes a complexity measure of the hypothesis class and $N$ is the number of training samples. Since SINO achieves low training error with substantially fewer parameters than other neural operators, it induces a less complex hypothesis class, which intuitively explains its strong performance in low-data regimes.

## 5 EXPERIMENTS

Given 2-5 observed trajectories, we evaluate SINO against baselines across various 2D and 3D scenarios without any known PDE terms. We also assess its OOD generalization and conduct ablation studies. **Source code will be released upon acceptance.**

### 5.1 DATASETS AND BASELINE MODELS

**Datasets.** We evaluate SINO on three PDE classes, each with distinct computational challenges.

**(1) Kuramoto-Sivashinsky Equation (KSE).** We consider the 2D KSE (Jayaprakash et al., 1993):

$$\partial_t u = -\nabla^2 u - \nabla^4 u - 0.5|\nabla u|^2, \quad \mathbf{x} \in [0, 12\pi)^2, \ t \in [0, 5]. \tag{7}$$

The KSE is known for its stiffness due to a biharmonic term $\nabla^4 u$ and the emergence of spatiotemporal chaos. We use this PDE to evaluate the performance of SINO in handling high-order derivatives and denote the corresponding experiment as E1.

**(2) Navier-Stokes Equation (NSE).** To examine globally coupled nonlinear dynamics and assess extrapolation ability, we adopt the 2D incompressible NSE in vorticity form as follows:

$$\partial_t \omega = \nu \nabla^2 \omega - (\mathbf{u} \cdot \nabla)\omega + f, \quad \mathbf{x} \in [0, 1)^2, \ t \in [0, 15], \tag{8}$$

where $\omega$ is the vorticity and $\mathbf{u}$ is the velocity. NSE presents several challenges like multi-scale turbulence and nonlocal coupling between $\omega$ and $\mathbf{u}$. To assess performance under varying tasks, we consider two viscosities ($\nu \in \{10^{-4}, 10^{-5}\}$) and two forcings ($f_1(\mathbf{x}) = 0.1\cos(8\pi x_1)$, $f_2(\mathbf{x}) = 0.1\sqrt{2}\sin(2\pi(x_1 + x_2) + \frac{\pi}{4})$), yielding four setups (E2-E5). To evaluate extrapolation ability, we provide the first 10 s of data during training, while testing extends to 15 s.

**(3) Burgers' equation.** We consider Burgers' equation, a canonical model for nonlinear advection and shock formation:

$$\partial_t \mathbf{u} = 0.01\nabla^2 \mathbf{u} - (\mathbf{u} \cdot \nabla)\mathbf{u}, \quad \mathbf{x} \in [0, 2\pi)^d, \ t \in [0, 2], \tag{9}$$

where $d$ denotes the spatial dimension. In our experiments, we consider a standard 2D case ($d = 2$, E6), a 2D mixed-boundary case with Dirichlet conditions at the top and bottom and periodic conditions on the left and right ($d = 2$, E8), and a 3D case ($d = 3$, E7).

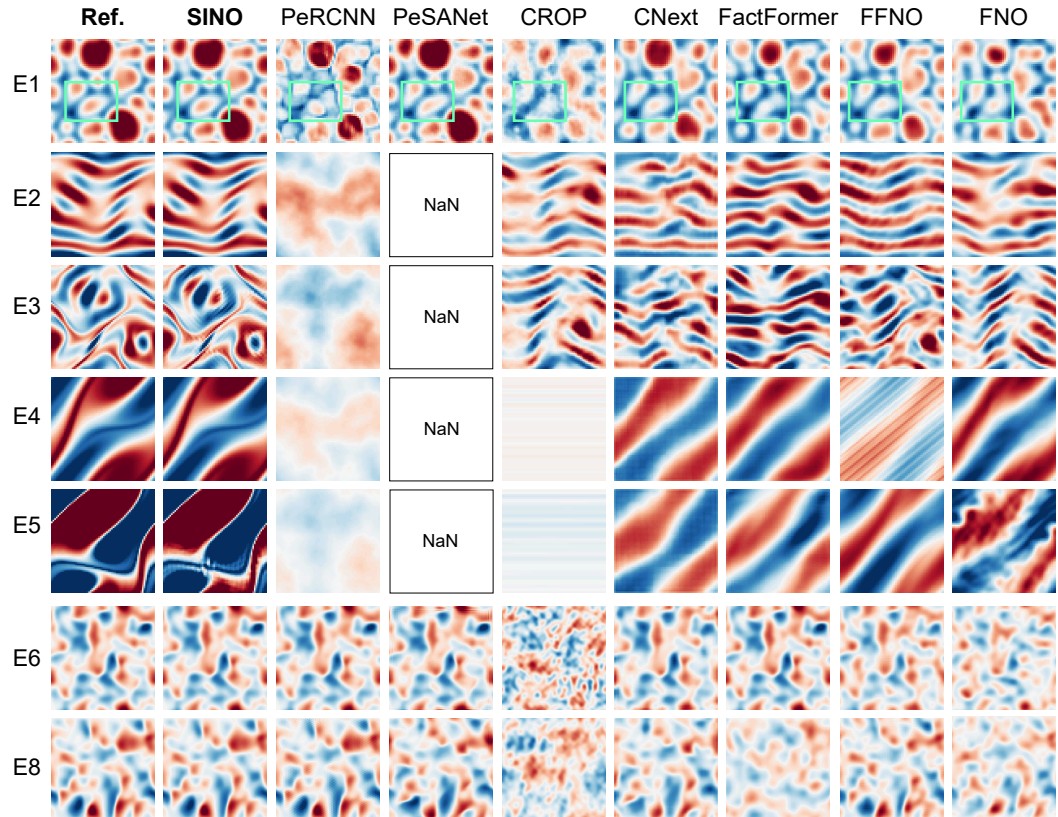

Figure 2: **Predicted snapshots of SINO and other baselines on 2D cases.** We show the final prediction plots for each method (E1-E6 and E8).

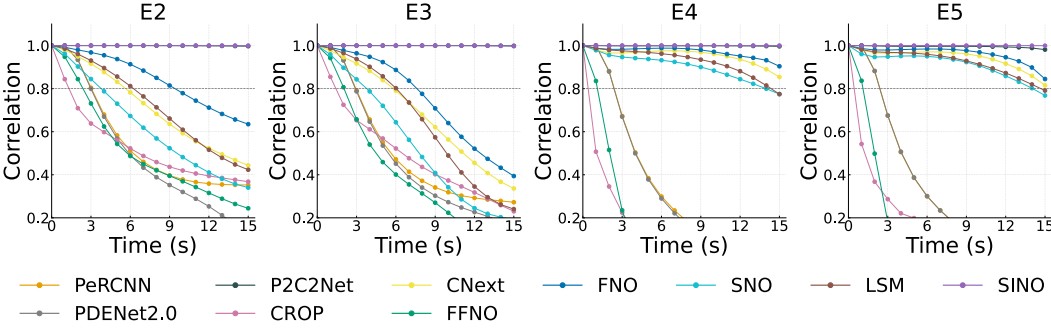

Figure 3: **Propagation curves for correlation across E2-E5 on NSEs.** Here, we provide the first 10 s of data during training, while testing extends to 15 s.

All datasets are simulated with high-resolution spectral solvers. We subsample 2-5 low-resolution trajectories per case for training. Dataset details are in Appendix 3.

**Baseline models.** We compare SINO against a broad set of data-driven and physics-encoded baselines. Data-driven operator learners include **FNO** (Li et al., 2021), **FFNO** (Tran et al., 2023), **FactFormer** (Li et al., 2024b), **CNext** (Liu et al., 2022; Ohana et al., 2024), **CROP** (Gao et al., 2025), **SNO** (Fanaskov & Oseledets, 2024), and **LSM** (Wu et al., 2023). We further include physics-encoded baselines such as **PeRCNN** (Rao et al., 2023), **PeSANet** (Wan et al., 2025), **PDENet 2.0** (Long et al., 2019), and **P²C²Net** (Wang et al., 2024c). Here, P²C²Net assumes full access to the governing physics and embeds a classical numerical solver into its architecture (FDM); we include it as a strong physics-informed baseline to highlight the advantage of SINO under our challenging settings. For fairness, all physics-encoded baselines are run with the same time-integration scheme as SINO, e.g., the RK4 scheme.

Table 1: **Relative $\ell_2$ error for seven cases.** The best results are shown in **bold**, and the second-best are underlined. NaN denotes 'Not a Number', and NA denotes 'Not Applicable'. # indicates methods that require full physical information and include a classical numerical solver inside.

| Model | KSE | NSE | | | | Burgers | | |
|---|---|---|---|---|---|---|---|---|
| | E1 | E2 $10^{-4}, f_1$ | E3 $10^{-5}, f_1$ | E4 $10^{-4}, f_2$ | E5 $10^{-5}, f_2$ | E6 2D | E7 3D | E8 MixedBC |
| **PeRCNN** (Rao et al., 2023) | 0.4027 | 0.8161 | 0.8341 | 0.9136 | 0.9127 | 0.0174 | 0.9530 | 0.0170 |
| **PeSANet** (Wan et al., 2025) | 0.0833 | NaN | NaN | NaN | NaN | 0.0974 | NA | 0.0967 |
| **CROP** (Gao et al., 2025) | 0.4541 | 0.9925 | 1.0021 | 0.9982 | 0.9939 | 0.9864 | 1.0826 | 0.9833 |
| **CNext** (Ohana et al., 2024) | 0.2256 | 0.7060 | 0.7388 | 0.2877 | 0.3153 | 0.2007 | 0.2098 | 0.2007 |
| **FactFormer** (Li et al., 2024b) | 0.2962 | 0.8102 | 0.9249 | 0.2879 | 0.3287 | 0.5446 | 0.4842 | 0.5908 |
| **FFNO** (Tran et al., 2023) | 0.2136 | 0.9484 | 1.0218 | 1.1791 | 1.1565 | 0.2374 | 0.1678 | 0.2837 |
| **FNO** (Li et al., 2021) | 0.3179 | 0.5210 | 0.6253 | 0.2282 | 0.2574 | 0.4157 | 0.9032 | 0.4495 |
| **SNO** (Fanaskov & Oseledets, 2024) | 0.6776 | 0.7774 | 0.8446 | 0.3526 | 0.4009 | 1.0001 | NA | 1.0001 |
| **LSM** (Wu et al., 2023) | 0.2652 | 0.6845 | 0.7293 | 0.3470 | 0.3664 | 0.1018 | NA | 0.0962 |
| **PDENet 2.0** (Long et al., 2019) | 0.4703 | 0.9587 | 0.9737 | 0.9207 | 1.1070 | 0.2254 | NA | 0.2210 |
| **P$^2$C$^2$Net$^\#$** (Wang et al., 2024c) | 0.3192 | 0.0346 | **0.0215** | 0.0526 | 0.1015 | 0.0110 | NA | 0.0138 |
| **SINO (ours)** | **0.0122** | **0.0110** | 0.0303 | **0.0031** | **0.0171** | **0.0008** | **0.0097** | **0.0002** |

## 5.2 MAIN RESULTS

We herein present the experimental results, with additional results provided in Appendix D.

**KSE (E1).** For KSE cases, PeRCNN suffers from discretization error when approximating fourth-order derivatives on coarse grids due to its FD-based stencils. A similar discretization error is observed for PDENet 2.0 and P$^2$C$^2$Net, whose internal finite-difference solvers are not accurate enough for low-resolution KSE. However, its error remains $6.8\times$ larger than that of SINO (Table 1). Furthermore, data-driven methods miss the major peaks of the solution field, resulting in inaccurate predictions (Fig. 1).

**NSE (E2-E5).** For the NSE cases, the challenge lies not only in capturing the nonlinear coupling between velocity and vorticity without explicit priors, but also in extrapolating beyond the training horizon. SINO maintains high correlation with the ground-truth over long rollouts (Fig. 3) and preserves coherent vortex structures even at the final timestep (Fig. 1). By contrast, PeRCNN and PeSANet collapse in the absence of sufficient physical priors, and PDENet 2.0 also suffers from large errors at long times, while FNO and CNext together with the other data-driven baselines, including SNO and LSM, perform best among the data-driven baselines but still cannot resolve vortices across multiple scales. Quantitatively, SINO improves upon these methods by 1–2 orders of magnitude (Table 1). It is also comparable to or better than P$^2$C$^2$Net, which assumes full access to the governing equations and relies on an internal finite-difference solver. Furthermore, we find that certain feature maps from SINO's $\Pi$-block and linear block align closely with ground-truth operators in NSEs (e.g., $\partial_x\omega$, $\Delta\omega$), providing strong evidence that SINO implicitly learns physically meaningful differential operators within the spectral learning block (Fig. 4).

**Burgers (E6-E8).** For the 2D Burgers' equation, we observe that PeRCNN and PeSANet perform reasonably well due to their local operator representations, which align naturally with the equation's local advection-diffusion structure. This observation also holds on the new mixed-boundary case E8, where the solution satisfies Dirichlet conditions at the top and bottom and periodic conditions in the horizontal direction, showing that these operator-based baselines remain competitive under more complex boundary conditions. Nevertheless, thanks to the spectral accuracy inherited from its design, SINO achieves lower error than the best baseline on E6 (0.0008 vs. 0.0110) and nearly two orders of magnitude lower error on the

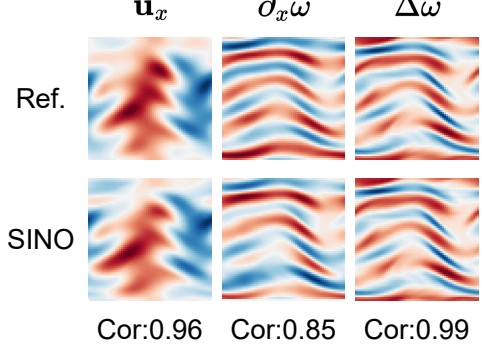

**u**$_x$  $\partial_x\omega$  $\Delta\omega$

Ref.

SINO

Cor:0.96  Cor:0.85  Cor:0.99

Figure 4: **Feature maps learned by SINO.** Feature maps from $\Pi$ / linear block align with ground-truth operators (e.g., $\partial_x\omega$, $\Delta\omega$) on E2.

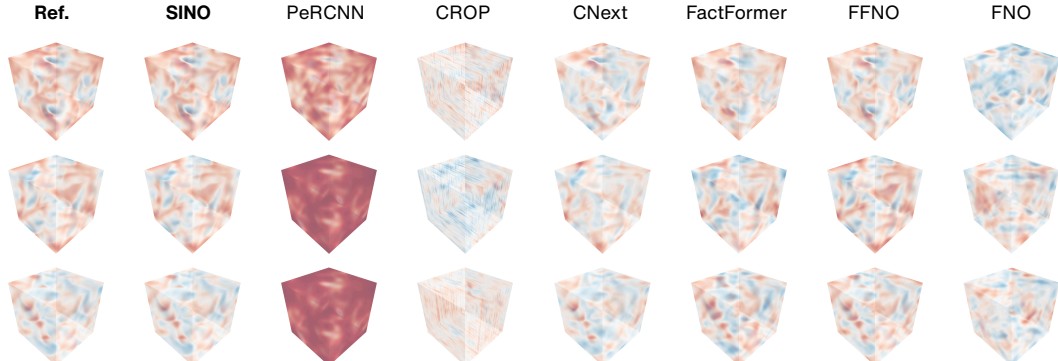

Figure 5: **Predicted snapshots of SINO and other baselines on 3D Burgers cases (E7).** The three rows denote the three components of $\mathbf{u} = [u, v, w]$, respectively.

mixed-boundary case E8 (0.0002 vs. 0.0138) (Table 1). In the 3D case, only FFNO and SINO are able to preserve all three velocity components in close agreement with the reference (Fig. 5). However, SINO better captures small-scale shocks and dissipation, producing solutions that remain accurate and physically consistent across all three components.

### 5.3 OOD GENERALIZATION TO DIFFERENT ICS

OOD generalization is a fundamental open challenge in operator learning, as models trained on limited distributions often fail when confronted with novel initial conditions or other new physics (Wang et al., 2024a; Zhang et al., 2025b). To further evaluate SINO's generalization ability, we compare it with data-driven baselines on the NSE case (E2). SINO is trained with only **5** trajectories, while baseline methods (including CNext, FactFormer, and FNO) are trained with **1000** trajectories. We test on four types of initial conditions: IC0, sampled from the same Gaussian random field distribution as the training set (in-distribution), and three OOD conditions, including IC1 (star), IC2 (smiley face), and IC3 (the pattern 'AI'). As shown in Fig. 6, all models perform reasonably well on the in-distribution case (IC0), but baselines fail to produce meaningful predictions on OOD cases, collapsing into unrealistic fields. In contrast, SINO accurately tracks the long-term dynamics for all OOD conditions, with results closely aligned with ground-truth even at the final timestep. This demonstrates that SINO does not rely on data memorization but genuinely learns the underlying operator mapping functions to functions. Additional rollout trajectories are provided in Appendix D.10.

### 5.4 ABLATION STUDIES

We conduct ablation experiments on three systems to evaluate the contributions of individual modules in Table 2. The experiments are conducted on three tasks: KSE (E1), NSE (E4), and Burgers (E6). Replacing the $\Pi$-block with a linear combination (**SINO \ $\Pi$-block**) weakens overall performance, particularly for the NSE, which relies on crucial nonlinear terms. Removing the low-pass filter (**SINO \ Filter**) introduces instability in the model's rollout, with aliasing effects becoming most pronounced in the NSE sys-

Table 2: **Relative $\ell_2$ error for ablation studies.**

| Model | KSE | NSE | Burgers |
|---|---|---|---|
| **SINO \ $\Pi$-block** | 0.4911 | 0.4028 | 0.2287 |
| **SINO \ Filter** | 0.0203 | NaN | 0.0063 |
| **SINO \ Freq2Vec** | NaN | NaN | 0.0093 |
| **SINO \ Linear** | NaN | NaN | NaN |
| **SINO \ RK4** | 0.0287 | NaN | 0.0049 |
| **SINO** | 0.0122 | 0.0031 | 0.0008 |

tem. Additionally, substituting the Freq2Vec module with a learnable vector (**SINO \ Freq2Vec**) or removing the linear block (**SINO \ Linear**) also significantly degraded the model's performance. Replacing the RK4 scheme with an Euler integrator (**SINO \ RK4**) further reduces accuracy and long-term stability, confirming that high-order temporal integration is critical for preserving accurate dynamics over extended rollouts. These experiments demonstrate that every component of the SINO design is indispensable, and it is their joint effect that endows SINO with high accuracy and strong generalization performance.

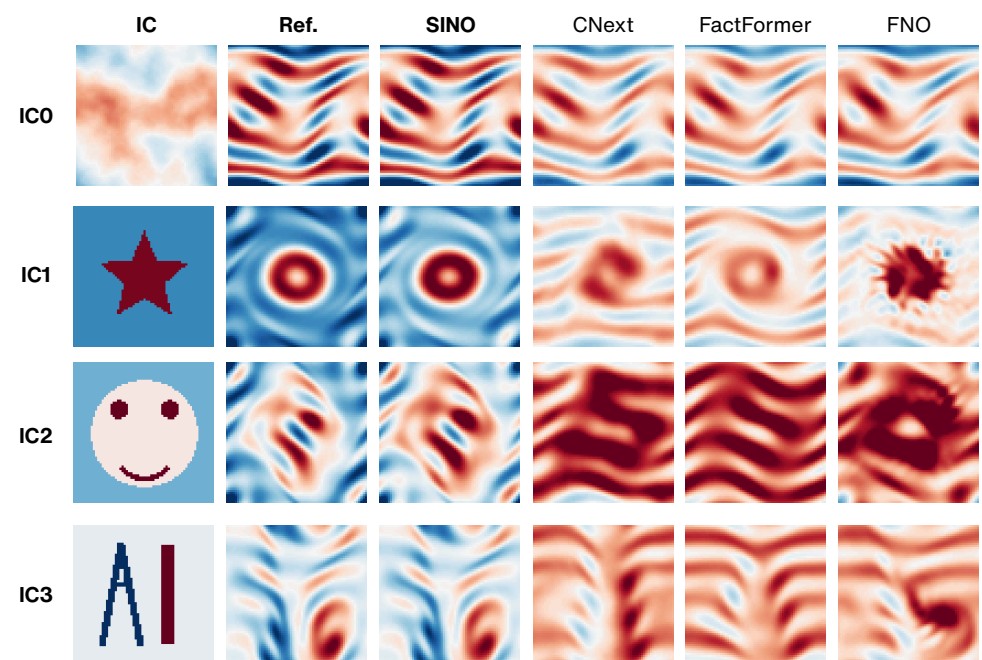

Figure 6: **In-distribution vs. OOD generalization on NSEs (E2).** Comparison between SINO trained with only **5** trajectories and data-driven baselines trained with **1000** trajectories. IC0 is sampled from the same distribution as the training set (in-distribution), while IC1-IC3 are sampled from OOD initial conditions. Shown are the vorticity fields after 15 s of evolution.

## 5.5 ADDITIONAL NUMERICAL RESULTS

In Appendix D, we provide further discussions and experiments beyond the main results. These include: (1) analysis of how training data quantity affects SINO's performance, (2) parameter sensitivity studies, (3) evaluations of zero-shot discretization invariance, (4) inference speed benchmarks, (5) evolution of training loss and validation error, (6) kinetic energy spectrum analysis, (7) consistency of learned operators, (8) sensitivity to de-aliasing ratios, (9) sensitivity to step size, and (10) additional trajectory predictions of physical fields for more comprehensive visualization.

## 6 CONCLUSION

We presented SINO, a spectral-inspired neural operator designed to model PDE-governed systems from limited data without requiring explicit knowledge of governing equations. By leveraging frequency-domain embeddings through the proposed Freq2Vec module and incorporating multiplicative interactions via the Π-block with de-aliasing, SINO provides a compact yet expressive architecture that balances accuracy and generalization. Theoretically, we established its universal approximation ability for a broad class of PDE operators. Empirically, SINO consistently outperforms both data-driven and physics-encoded baselines across diverse benchmarks, achieving 1-2 orders of magnitude lower error while maintaining robust OOD generalization. Our current study mainly focuses on Cartesian grids, and one promising direction is to extend SINO beyond regular domains to more complex geometries, which may be achieved through coordinate transformations that map irregular domains into regular ones, or by selecting appropriate basis functions tailored to the corresponding geometry.

## ETHICS STATEMENT

This work focuses on developing a general neural operator framework (SINO) for learning PDE-governed dynamics from scarce data. Our study does not involve human subjects, sensitive personal data, or applications with immediate societal harm.

## REPRODUCIBILITY STATEMENT

We have made extensive efforts to ensure the reproducibility of our results. The model architecture and training settings are described in Section 3, with dataset generation details provided in Appendix C.1. Theorem proofs are included in Appendix B, and extended experimental results are provided in Appendix D. Our model follows standard machine learning practices, where all hyperparameters are tuned based on validation performance, ensuring fair and consistent model selection. **Source codes will be released publicly upon acceptance.**

## LLM USAGE STATEMENT

Large Language Models (LLMs) such as ChatGPT were used in this work for language editing (e.g., grammar checking, improving readability, and converting text to LaTeX format) and for generating initial code templates in Python. All scientific ideas, model designs, theoretical results, experiments, and analysis were conceived, implemented, and validated solely by the authors.

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

APPENDIX OVERVIEW

## A  PROOF

## B  PROOF OF THEOREM 1

**Theorem 1 (Approximation ability of SINO)**   Let $\Omega = [0,1)^d$ denote the $d$-torus and define the set $\mathcal{U} := \left\{ \mathbf{v} : \Omega \to \mathbb{R}^C \text{ periodic, bandlimited to } \|\mathbf{k}\|_\infty \leq k_{\max}, \ \|\mathbf{v}\|_{L^2} \leq B \right\}$ for some $B > 0$. Let $\mathbf{u} : \Omega \to \mathbb{R}^C$ be any function with $\mathbf{u} \in \mathcal{U}$. Suppose the right-hand side (RHS) of the PDE can be expressed as

$$\partial_t \mathbf{u} = \mathcal{R}(\mathbf{u}) := \sum_{j=1}^{J} \prod_{i=1}^{n_j} \mathcal{D}_{j,i} \mathbf{u},$$

where each $\mathcal{D}_{j,i}$ is a linear differential operator of finite order. Then for any $\epsilon > 0$, there exists a SINO model $\mathcal{S}_\theta$ such that

$$\sup_{\mathbf{u} \in \mathcal{U}} \big\| \mathcal{S}_\theta(\mathbf{u}) - \mathcal{R}(\mathbf{u}) \big\|_{L^2} < \epsilon.$$

**Proof 1** *Let us define a truncated version of* $\mathbf{u}$ *in the Fourier domain:*

$$\mathbf{u}_{k_{\max}}(\mathbf{x}) = \sum_{\|\mathbf{k}\|_\infty \leq k_{\max}} \hat{\mathbf{u}}(\mathbf{k}) e^{2\pi i \mathbf{k} \cdot \mathbf{x}}.$$

*Since* $\mathbf{u}$ *is bandlimited, we have* $\mathbf{u} = \mathbf{u}_{k_{\max}}$.

*Each linear differential operator $\mathcal{D}_{j,i}$ corresponds to a spectral multiplier:*

$$\mathcal{D}_{j,i}\mathbf{u}(\mathbf{x}) = \sum_{\|\mathbf{k}\|_\infty \le k_{\max}} \psi_{j,i}(\mathbf{k})\hat{\mathbf{u}}(\mathbf{k})e^{2\pi i\mathbf{k}\cdot\mathbf{x}},$$

*where $\psi_{j,i}(\mathbf{k})$ is a function of $\mathbf{k}$.*

*Let $\widehat{\mathcal{D}}_{j,i}$ denote the learned spectral operator in SINO. By the universal approximation theorem for neural networks, we can construct a neural network approximator $\hat{\psi}_{j,i}(\mathbf{k})$ for any $\delta > 0$, such that*

$$\sup_{\|\mathbf{k}\|_\infty \le k_{\max}} \left|\hat{\psi}_{j,i}(\mathbf{k}) - \psi_{j,i}(\mathbf{k})\right| < \delta.$$

*This implies element-wise approximation of Fourier multipliers, and consequently*

$$\left\|\widehat{\mathcal{D}}_{j,i}\mathbf{u} - \mathcal{D}_{j,i}\mathbf{u}\right\|_{L^2} \le C_{j,i}(B, k_{\max})\,\delta.$$

*Now, consider the nonlinear composition $\prod_{i=1}^{n_j} \mathcal{D}_{j,i}\mathbf{u}$ approximated by $\prod_{i=1}^{n_j} \widehat{\mathcal{D}}_{j,i}\mathbf{u}$. Since bandlimited functions are smooth and satisfy uniform bounds, multiplication is continuous in $L^2$, implying that the error of the product can be controlled linearly in $\delta$:, i.e.,*

$$\left\|\prod_{i=1}^{n_j} \widehat{\mathcal{D}}_{j,i}\mathbf{u} - \prod_{i=1}^{n_j} \mathcal{D}_{j,i}\mathbf{u}\right\|_{L^2} \le C_j(B, k_{\max}, n_j)\,\delta,$$

*for some constant $C_j(B, k_{\max}, n_j)$ independent of the particular $\mathbf{u}$.*

*Aggregating over all $j = 1, \ldots, J$ gives*

$$\|\mathcal{S}_\theta(\mathbf{u}) - \mathcal{R}(\mathbf{u})\|_{L^2} \le \sum_{j=1}^{J} C_j(B, k_{\max}, n_j)\,\delta = C'(B, k_{\max})\,\delta.$$

*Therefore, for any prescribed $\epsilon > 0$, choosing $\delta < \epsilon/C'(B, k_{\max})$ and increasing the neural network capacity accordingly, we ensure*

$$\sup_{\mathbf{u}\in\mathcal{U}} \|\mathcal{S}_\theta(\mathbf{u}) - \mathcal{R}(\mathbf{u})\|_{L^2} < \epsilon.\blacksquare$$

**On unbounded Fourier bandwidth.** The theorem assumes that the input function $\mathbf{u}$ is bandlimited with $\|\mathbf{k}\|_\infty \le k_{\max}$. In practice, many physical fields are not exactly bandlimited. However, if $\mathbf{u} \in C^m(\Omega)$, then its Fourier coefficients decay at the rate

$$|\hat{\mathbf{u}}(\mathbf{k})| \lesssim \frac{1}{\|\mathbf{k}\|^m},$$

which implies that the tail $\|\mathbf{u} - \mathbf{u}_{k_{\max}}\|_{L^2}$ decays as $O(k_{\max}^{-(m-d/2)})$. Thus, the total approximation error of SINO in this case can be bounded by

$$\|\mathcal{S}_\theta(\mathbf{u}) - \mathcal{R}(\mathbf{u})\|_{L^2} \le \epsilon + C_2 k_{\max}^{-(m-d/2)}.$$

By choosing a sufficiently large $k_{\max}$ and network capacity, this bound can still be made arbitrarily small.

**Supported PDE types.** The class of PDEs covered by this theorem includes a broad family of physical systems, as long as the RHS can be expressed as a sum of products of linear differential operators. This encompasses reaction-diffusion equations, convection-diffusion equations, the incompressible Navier-Stokes equations, and other general polynomial-type PDEs.

Table 3: **Summary of experimental settings for different cases.** For the NSE case, models were trained on trajectories spanning 10 s and evaluated on prediction rollouts of 15 s. "Gen." denotes the resolution/time step used for generating high-fidelity data with spectral solvers. "DOL" and "POL" denotes the resolution/time step used for data-driven and physics-encoded operator learning.

| Setting | KSE | NSE | 2D Burgers | 3D Burgers |
|---|---|---|---|---|
| Numerical Method | spectral method | spectral method | spectral method | spectral method |
| Spatial Domain | $[0, 12\pi)^2$ | $[0, 1)^2$ | $[0, 2\pi)^2$ | $[0, 2\pi)^3$ |
| Temporal Domain | $[0, 5]$ | $[0, 10/15]$ | $[0, 2]$ | $[0, 5]$ |
| Training Trajectories | 2 | 5 | 5 | 5 |
| Validation Trajectories | 2 | 2 | 2 | 2 |
| Test Trajectories | 5 | 5 | 5 | 5 |
| Gen. Grid / $\Delta t$ | $108^2, 1\times10^{-4}$ | $256^2, 1\times10^{-4}$ | $512^2, 1\times10^{-3}$ | $128^3, 5\times10^{-3}$ |
| DOL Grid / $\Delta t$ | $54^2, 1$ | $64^2, 1$ | $128^2, 0.2$ | $64^3, 0.5$ |
| POL Grid / $\Delta t$ | $54^2, 1\times10^{-3}$ | $64^2, 5\times10^{-3}$ | $128^2, 5\times10^{-3}$ | $64^3, 5\times10^{-2}$ |

## C  EXPERIMENTAL DETAILS

### C.1  DATA GENERATION

Initial conditions for all PDE systems were sampled from a Gaussian Random Field (GRF), following the standard practice introduced in the classical FNO work (Li et al., 2021). We target the data-scarce scenario: each experiment includes only 2-5 trajectories in the training set, as summarized in Table 3. The table specifies the numerical scheme used for data generation, the spatial and temporal domains, and the number of training and testing trajectories for each PDE case. For the NSE case, training trajectories span 10 s, while evaluation involves rollout prediction over 15 s, as indicated in the temporal domain row of Table 3. Numerical solutions were obtained using a $2/3$ dealiased pseudo-spectral method with periodic boundary conditions, implemented in Python with PyTorch, consistent with prior classical papers (Li et al., 2021; Tran et al., 2023). In particular, the NSE dataset was generated using an open-source implementation from (Li et al., 2021). To ensure reproducibility, we fixed random seeds for dataset splitting: seed 0 for training, seed 1 for validation, and seed 2 for testing.

### C.2  BASELINE MODELS

We provide detailed introductions to the baseline models used for comparison with SINO. These baselines include data-driven approaches such as FNO (Li et al., 2021), FFNO (Tran et al., 2023), FactFormer (Li et al., 2024b), CNext (Liu et al., 2022; Ohana et al., 2024) and CROP (Gao et al., 2025), as well as physics-encoded models including PeRCNN (Rao et al., 2023) and PeSANet (Wan et al., 2025). Details of baseline models are provided as follows:

**Fourier Neural Operator (FNO)** (Li et al., 2021). FNO is one of the most classical data-driven neural operator model that captures features in the frequency domain. Its ability to capture information in the frequency domain allows it to effectively utilize global information, and it also possesses a degree of resolution invariance.

For hyperparameter tuning, we conducted a grid search over the number of channels (12, 36, 64) and Fourier modes (12, 16), and selected the best configuration according to the performance on the validation set.

**Factorized Fourier Neural Operator (FFNO)** (Tran et al., 2023). FFNO is an improved neural operator model based on FNO, and its core involves the introduction of factorized Fourier representation. This factorization method and improved network structure allow FFNO to employ deeper network architectures and demonstrate performance superior to standard FNO in simulating various partial differential equations.

For hyperparameter tuning, we performed a grid search over the number of channels (16, 32) and Fourier modes (12, 16), selecting the best setting based on validation performance.

**FactFormer** (Li et al., 2024b). FactFormer is a transformer-based model designed for multi-dimensional settings that leverages an axial factorized kernel integral, implemented via a learnable projection operator that decomposes the input function into one-dimensional sub-functions.

For hyperparameter tuning, we searched over the hidden dimension (64, 128, 256) and network depth (4, 6), and selected the optimal configuration according to the validation set.

**CNext** (Liu et al., 2022; Ohana et al., 2024). CNext is a family of modernized ConvNet models, inspired by the design principles of Vision Transformers (ViTs) while retaining the efficiency and simplicity of convolutional architectures. In the latest benchmark study (Ohana et al., 2024), CNext achieved SOTA results across a wide range of PDE learning tasks.

For hyperparameter tuning, we conducted a grid search over the network width (16, 32, 64) and number of layers (2, 3, 4), and selected the best configuration according to validation performance.

**Cross-Resolution Operator-learning Pipeline (CROP)** (Gao et al., 2025). CROP is a method proposed to address problems with generalization and discretization mismatch error in existing neural operators across different data resolutions. Equipped with a specific pipeline design, CROP is able to eliminate aliasing effects and discretization mismatch errors, thus enabling efficient learning and inference across different resolutions.

For hyperparameter tuning, we conducted a grid search over the number of channels (8, 12, 36) and Fourier modes (12, 16), and selected the best configuration according to validation performance.

**Physics-embedded Recurrent-Convolutional Neural Network (PeRCNN)** (Rao et al., 2023). PeRCNN is a physics-encoded learning methodology that directly embeds physical laws into the neural network architecture. It employs multiple parallel convolutional neural network and leverages feature map multiplication to simulate polynomial equations, thereby enhancing the model's extrapolation and generalization capabilities.

For hyperparameter tuning, we performed a grid search over the network width (32, 64, 128) and input kernel size (5, 7), selecting the best configuration based on validation performance.

**Physics-encoded Spectral Attention Network (PeSANet)** (Wan et al., 2025). PeSANet includes a physics-encoded block for approximating local differential operators and a spectral-enhanced block which, combined with spectral attention, captures global features in the frequency domain, allowing it to perform excellently, especially in long-term forecasting accuracy, under scarce data and incomplete physical priors.

For hyperparameter tuning, we searched over the network width (32, 64, 128) and input kernel size (5, 7), and selected the optimal configuration according to validation performance.

For the proposed **SINO**, we conducted a grid search over the number of channels (16, 32, 64) and the dimension of $\psi(\mathbf{k})$ in the Freq2Vec (4, 6, 8), selecting the best configuration according to validation performance.

Our model follows standard machine learning practices, where all hyperparameters are tuned based on validation performance with fixed seed, ensuring fair and consistent model selection. The parameter sizes of all baseline models and SINO are summarized in Table 4.

### C.3 METRICS

To assess the performance of our proposed method, we utilize two evaluation metrics: relative $\ell_2$ error and correlation. The relative $\ell_2$ error measures the ratio of the $\ell_2$ norm of the error vector to that of the ground-truth vector, providing a dimensionless measure of the prediction error relative to the true scale. Correlation is quantified using the Pearson correlation coefficient (PCC), which measures the linear dependence between predicted and ground-truth solutions over time.

The definitions are as follows:

$$
\begin{aligned}
\text{Relative } \ell_2 \text{ Error:} \quad & \frac{\|\mathbf{y} - \tilde{\mathbf{y}}\|_2}{\|\mathbf{y}\|_2} = \sqrt{\frac{\sum_{i=1}^{n}(y_i - \tilde{y}_i)^2}{\sum_{i=1}^{n} y_i^2}}, \\
\text{Correlation (PCC):} \quad & \text{PCC}(\mathbf{y}, \tilde{\mathbf{y}}) = \frac{\text{Cov}(\mathbf{y}, \tilde{\mathbf{y}})}{\sigma_{\mathbf{y}} \sigma_{\tilde{\mathbf{y}}}}.
\end{aligned} \tag{10}
$$

Table 4: **Parameter counts of different models.** NaN denotes 'Not a Number', and NA denotes 'Not Applicable'.

| Model | KSE | NSE | | | | Burgers | | |
|---|---|---|---|---|---|---|---|---|
| | E1 | E2 $10^{-4}, f_1$ | E3 $10^{-5}, f_1$ | E4 $10^{-4}, f_2$ | E5 $10^{-5}, f_2$ | E6 2D | E7 3D | E8 MixedBC |
| **FNO** (Li et al., 2021) | 0.59M | 0.33M | 9.46M | 0.33M | 3.00M | 9.46M | 127.41M | 16.8M |
| **FFNO** (Tran et al., 2023) | 17.85M | 0.48M | 0.48M | 13.65M | 0.48M | 4.33M | 62375 | 4.32M |
| **FactFormer** (Li et al., 2024b) | 11.26M | 11.26M | 2.95M | 1.81M | 2.95M | 0.65M | 0.30M | 0.133M |
| **CNext** (Ohana et al., 2024) | 50977 | 51.35M | 51.35M | 12.84M | 0.84M | 0.22M | 4.16M | 0.21M |
| **CROP** (Gao et al., 2025) | 2.76M | 0.56M | 0.56M | 0.56M | 0.56M | 2.76M | 22.13M | 0.13M |
| **PeSANet** (Wan et al., 2025) | 89162 | NaN | NaN | NaN | NaN | 89262 | NA | 89262 |
| **PeRCNN** (Rao et al., 2023) | 10116 | 10116 | 10116 | 10116 | 10116 | 19716 | 1924 | 76292 |
| **SNO** (Fanaskov & Oseledets, 2024) | 81998 | 0.15M | 0.28M | 0.28M | 0.15M | 2.23M | NA | 2.23M |
| **LSM** (Wu et al., 2023) | 19.18M | 19.19M | 43.12M | 19.19M | 43.12M | 43.12M | NA | 43.12M |
| **PDENet 2.0** (Long et al., 2019) | 252 | 252 | 156 | 252 | 156 | 1424 | NA | 320 |
| **P$^2$C$^2$Net** (Wang et al., 2024c) | 2532 | 38.91M | 38.91M | 38.91M | 38.91M | 4932 | NA | 4932 |
| **SINO (ours)** | 1708 | 9278 | 9278 | 9278 | 9278 | 11133 | 2951 | 13651 |

Here, $n$ denotes the number of evaluation points, $y_i$ and $\tilde{y}_i$ are the $i$-th components of the ground-truth vector $\mathbf{y}$ and the predicted vector $\tilde{\mathbf{y}}$, respectively, Cov is the covariance, and $\sigma$ denotes the standard deviation.

## C.4 Training Details

All experiments were conducted on a single NVIDIA A100 GPU (80GB) with an Intel(R) Xeon(R) Platinum 8380 CPU (2.30GHz, 64 cores). For 2D cases, models were trained for 20000 iterations, while for 3D cases, 5000 iterations were used. We adopt a OneCycle learning rate scheduler with a maximum learning rate selected from $\{0.01, 0.001\}$, scheduled over the entire training horizon.

During training, we randomly sample a starting position along each trajectory. From this state, the model first evolves forward $n$ steps without gradients (warm-up), where $n$ is randomly sampled from $\{0, \ldots, n_1\}$ at each iteration, and then predicts the following $n_2$ steps with gradients, which are compared against the ground truth. For physics-encoded methods, we fix $n_1 = 4$ and $n_2 = 8$; for data-driven methods, due to their larger effective step size, we set $n_1 = 2$ and $n_2 = 2$. This scheme provides stable backpropagation and encourages the models to generalize across different rollout horizons (Brandstetter et al., 2022).

# D Additional Numerical Results

## D.1 Data Efficiency

To evaluate the data efficiency and scalability of SINO, we conduct experiments on KSE (E1) with varying numbers of training trajectories. The results in Table 5 reveal two key observations. First, SINO achieves strong performance even when trained on a single trajectory, with a relative $\ell_2$ error of 0.0430. This highlights the model's ability to capture complex dynamics from extremely limited data, which is crucial in data-scarce scientific domains. Second, SINO scales effectively with additional data: as the number of training trajectories increases from 1 to 8, the relative $\ell_2$ error decreases monotonically from 0.0430 to 0.0122. Although performance slightly saturates beyond 8 trajectories, the trend confirms that SINO can leverage more data to enhance predictive accuracy. Overall, these results demonstrate that SINO is both data-efficient and scalable: it delivers accurate predictions from minimal data while benefiting from larger training sets.

Table 5: **Relative $\ell_2$ error of SINO with varying training dataset sizes on KSE (E1).**

| Number of Trajectories | 1 | 2 | 4 | 8 | 16 |
|---|---|---|---|---|---|
| **Relative $\ell_2$ Error** | 0.0430 | 0.0285 | 0.0175 | 0.0122 | 0.0139 |

Figure 7: **Operator distillation.** The three stages of operator distillation include few-shot operator learning, synthetic data generation, and the training of SINO-FNO.

## D.2 HYPERPARAMETER SENSITIVITY ANALYSIS

Table 6 reports the sensitivity of SINO to two key hyperparameters: the channel size ($C \in \{64, 32, 16\}$) and the embedding dimension of $\psi(\mathbf{k})$ in the Freq2Vec module ($K \in \{8, 6, 4\}$). As expected, reducing either the number of channels or the embedding dimension degrades accuracy. For example, with $K = 8$, the relative $\ell_2$ error increases from $0.0109$ at $C = 64$ to $0.0593$ at $C = 16$, while reducing $K$ from 8 to 4 also significantly worsens performance. Nevertheless, across all tested settings, SINO maintains errors within a reasonable range, demonstrating robustness to hyperparameter choices.

Table 6: **Hyperparameter sensitivity.** Relative $\ell_2$ error for different configurations.

| Model | $C = 64$ | $C = 32$ | $C = 16$ |
|-------|----------|----------|----------|
| $K = 8$ | 0.0109 | 0.0245 | 0.0593 |
| $K = 6$ | 0.0392 | 0.0382 | 0.0457 |
| $K = 4$ | 0.1089 | 0.1216 | 0.1406 |

## D.3 ZERO-SHOT SUPER-RESOLUTION

We further evaluate the zero-shot super-resolution capability of SINO, where the model is trained on the lowest-resolution data and directly tested on higher resolutions without retraining. As summarized in Table 7, SINO trained on $36 \times 36$ (KSE) or $32 \times 32$ (NSE) generalizes robustly to finer grids. In the KSE case (E1), the relative $\ell_2$ error drops significantly from $0.1169$ at low resolution to $0.0297$ and $0.0283$ when applied to $54 \times 54$ and $108 \times 108$ grids, respectively. In the NSE case, the model achieves errors of $0.0523$, $0.0427$, and $0.0424$ when evaluated at $32 \times 32$, $64 \times 64$, and $128 \times 128$, respectively. These results highlight that SINO not only achieves discretization invariance but also benefits from higher-resolution inputs at inference, where the Freq2Vec module effectively extracts fine-scale spectral information to improve prediction accuracy.

Table 7: **Relative $\ell_2$ error for different resolutions.**

| Equation | Low | Medium | High |
|----------|-----|--------|------|
| KSE (E1) | 0.1169 | 0.0297 | 0.0283 |
| NSE (E4) | 0.0523 | 0.0427 | 0.0424 |

## D.4 INFERENCE SPEED AND OPERATOR DISTILLATION

In this section, we discuss the inference speed of SINO. Similar to other physics-encoded methods (McGreivy & Hakim, 2024), the speed of SINO is constrained by its intrinsic physical structure. Since the focus of this work is on modeling complex systems under physics-agnostic settings, computational efficiency is not the primary objective. Nevertheless, it is possible to alleviate this bottleneck with a simple yet effective strategy.

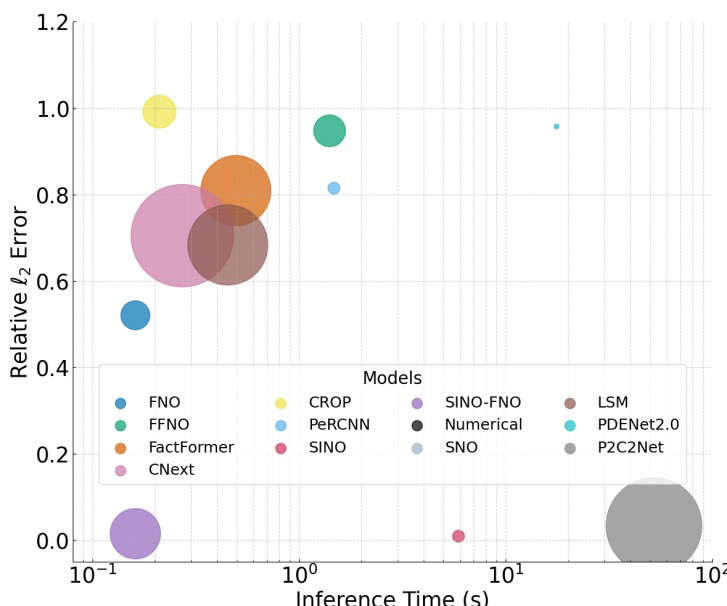

Figure 8: **Inference time vs. accuracy trade-off on NSE (E2). Each bubble denotes a model, where the $x$-axis shows inference time (s, log scale), the $y$-axis shows relative $\ell_2$ error, and the bubble size corresponds to parameter count.**

To this end, we introduce an *operator distillation* approach that transfers the knowledge of a fine-step-size SINO model into a coarse-step-size neural operator, termed **SINO-FNO** (Fig. 7). The procedure consists of three stages. **(i)** We first train a SINO model from limited observations and use the resulting surrogate operator as the teacher network. **(ii)** Equipped with this teacher, we sample new initial conditions from the underlying distribution and generate a large synthetic dataset by rolling out trajectories with the SINO teacher. **(iii)** We then train an FNO as the student network on this synthetic dataset. Importantly, in the distillation loss we supervise the FNO to predict system states over a much coarser time step than the fine resolution used in the teacher SINO.

It is worth noting that, in the physics-agnostic scenarios, numerical solvers cannot be used to generate training data. Instead, SINO itself serves as a proxy simulator. Since SINO achieves 1-2 orders of magnitude lower error than data-driven neural operators, it provides a sufficiently accurate teacher, making this distillation strategy both feasible and effective.

We conduct an inference time analysis of SINO and SINO-FNO, comparing them against baseline neural operators and the traditional spectral method (chosen as a representative numerical solver due to its accuracy and efficiency on regular domains (McGreivy & Hakim, 2024)). The analysis is performed on the NSE (E2) case. All models are tested on the same spatial resolution; step sizes are kept consistent across data-driven approaches (including SINO-FNO), while physics-aware models follow the same setup as SINO. For the spectral solver, we adopt the largest stable time step to ensure fairness.

As shown in Fig. 8, SINO achieves the lowest relative $\ell_2$ error among all neural operators, reaching accuracy comparable to the spectral method. Through operator distillation, SINO-FNO effectively reduces inference cost by more than two orders of magnitude (down to $\sim$0.2s), while retaining errors close to zero. This corresponds to a $\sim$36$\times$ speedup compared to the spectral method, with accuracy far surpassing purely data-driven baselines.

### D.5 EVOLUTION OF TRAINING LOSS AND VALIDATION ERROR

In this subsection, we present the training dynamics of SINO and other data-driven baselines on the NSE case (E3). As shown in Fig. 9, SINO maintains a consistent trend between training loss and validation error throughout the entire training process, both decreasing steadily without divergence.

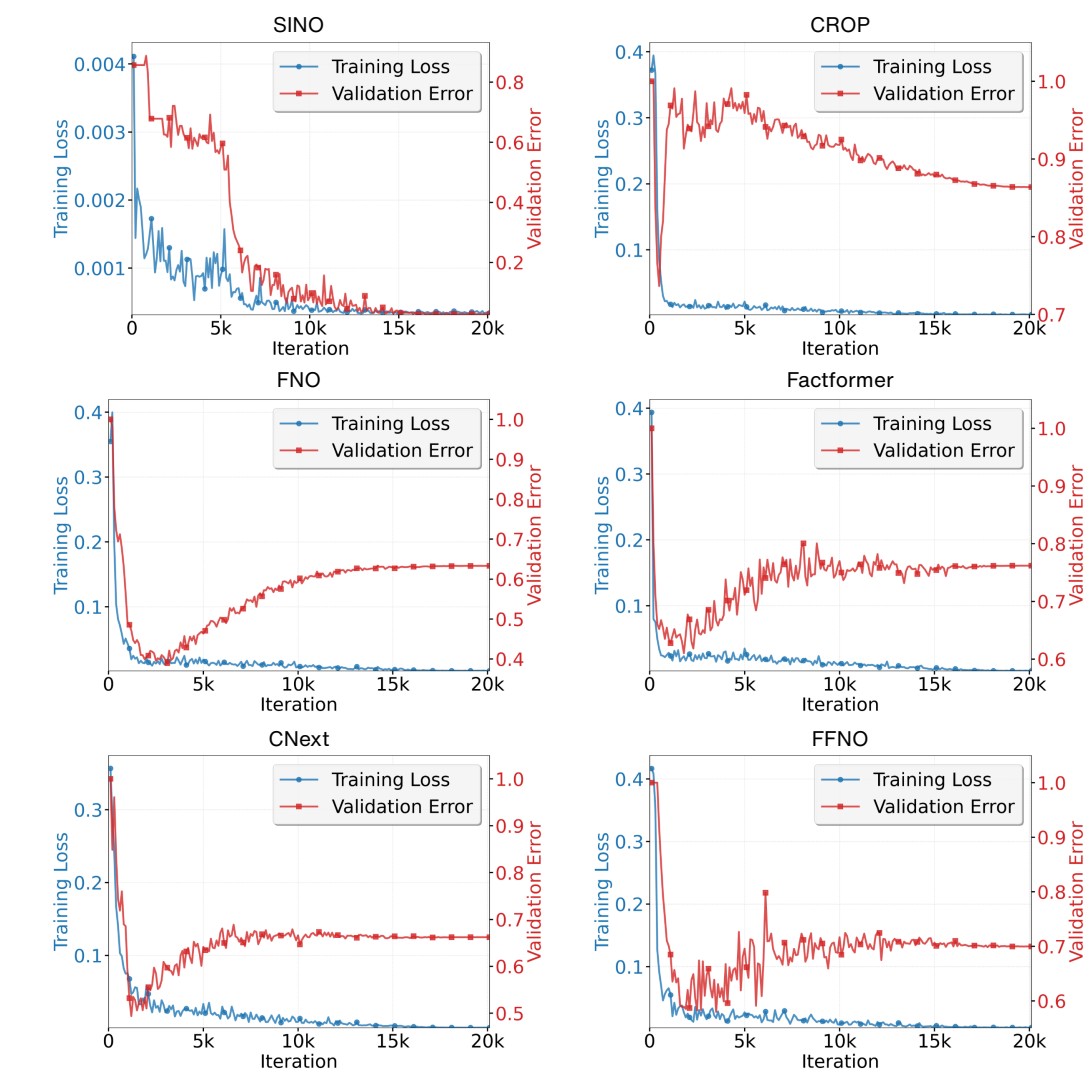

Figure 9: **Training and validation dynamics on NSE (E3).** Comparison of SINO with other data-driven baselines.

Table 8: Relative $\ell_2$ error comparison with different de-aliasing cutoff ratios.

| De-aliasing Cutoff | E2 | E3 | E4 | E5 |
|---|---|---|---|---|
| 2/3 (Original) | 0.0110 | 0.0303 | 0.0031 | 0.0171 |
| 0.8 | 0.0123 | 0.0298 | 0.0036 | 0.0185 |
| 0.5 | 0.0061 | 0.0474 | 0.0039 | 0.0261 |

In contrast, data-driven baselines suffer from severe overfitting: while their training loss continues to decrease, the validation error increases, indicating poor generalization. These results further corroborate our discussion in Section 4, highlighting that SINO achieves robust generalization rather than mere data memorization.

### D.6    KINETIC ENERGY SPECTRUM ANALYSIS

To further evaluate the physical fidelity of different spectral operator networks, we analyze the kinetic energy spectrum $E(\kappa)$ across multiple NSE cases (E2-E5) in Fig. 10. The spectral energy distribution provides insight into how each model captures multi-scale dynamics and energy trans-

Table 9: Performance (relative $\ell_2$ error) on E3 with varying RK4 step sizes.

| Step Size ($\Delta t$) | Relative $\ell_2$ Error |
|---|---|
| 0.005s (Original) | 0.0303 |
| 0.05s | 0.0223 |
| 0.5s | NaN |

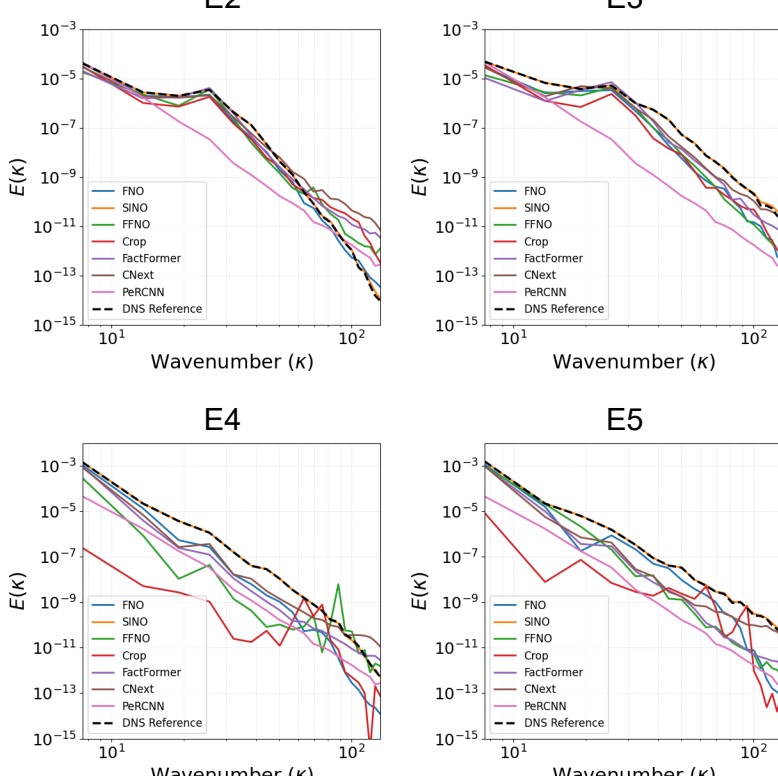

Figure 10: Spectral Energy Distribution of each on NSE cases (E2-E5).

fer mechanisms across wavenumbers. In all cases, SINO maintains close alignment with the DNS spectrum across the full range of $\kappa$, effectively preserving both large-scale and fine-scale structures.

### D.7 CONSISTENCY AND ROBUSTNESS OF LEARNED PHYSICAL OPERATORS

To further demonstrate the robustness of the proposed method, we extend the interpretability analysis to a broader range of physical scenarios. Specifically, we visualize the feature maps corresponding to the Laplacian operator ($\Delta\omega$) learned by SINO across datasets E2, E3, E4, and E5. As illustrated in Figure 11, the learned features consistently exhibit high alignment with the ground-truth operators, regardless of the underlying resolution or specific flow dynamics. These results provide strong evidence that SINO implicitly learns physically meaningful differential operators within the spectral learning block and maintains this interpretability robustly across different settings.

### D.8 SENSITIVITY TO DE-ALIASING CUTOFF RATIOS

To assess the impact of spectral de-aliasing, we vary the cutoff ratio from the default $2/3$ to $0.8$ and $0.5$ on the NSE cases E2–E5. As shown in Table 8, the relative $\ell_2$ errors change only moderately across all cases, indicating that SINO is numerically robust to the specific choice of cutoff ratio.

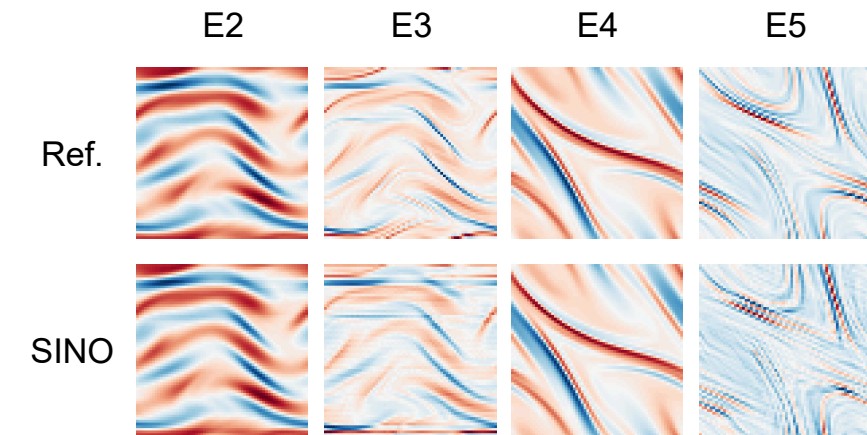

Figure 11: Visualization of learned feature maps corresponding to $\Delta\omega$ across datasets E2–E5.

We therefore adopt the classical 2/3 rule in all main experiments to align with standard practice in spectral methods.

### D.9 SENSITIVITY TO RK4 TIME STEP SIZE

To assess temporal stability, we vary the RK4 time step size ($\Delta t$) on E3 from the default 0.005s to 0.05s and 0.5s. As shown in Table 9, SINO maintains robust performance and even achieves slightly lower relative $\ell_2$ error at $\Delta t = 0.05$s, while an excessively large step size of 0.5s leads to divergence (NaN), as expected for explicit integrators. These results indicate that SINO is stable within a reasonable range of coarse time steps, and we adopt $\Delta t = 0.005$s in the main experiments as a conservative choice.

### D.10 ADDITIONAL ROLLOUT TRAJECTORIES

In this section, we provide additional rollout trajectories to complement the additional results in the main text. Fig. 12 illustrates long-term predictions on the NSE cases (E2-E5). Figs. 13-16 further examine generalization across different initial conditions, including both in-distribution (IC0) and out-of-distribution (IC1-IC3) scenarios. Moreover, we show the forecasting results of SNO, LSM, PDENet2.0 and P$^2$C$^2$Net in Fig. 17.

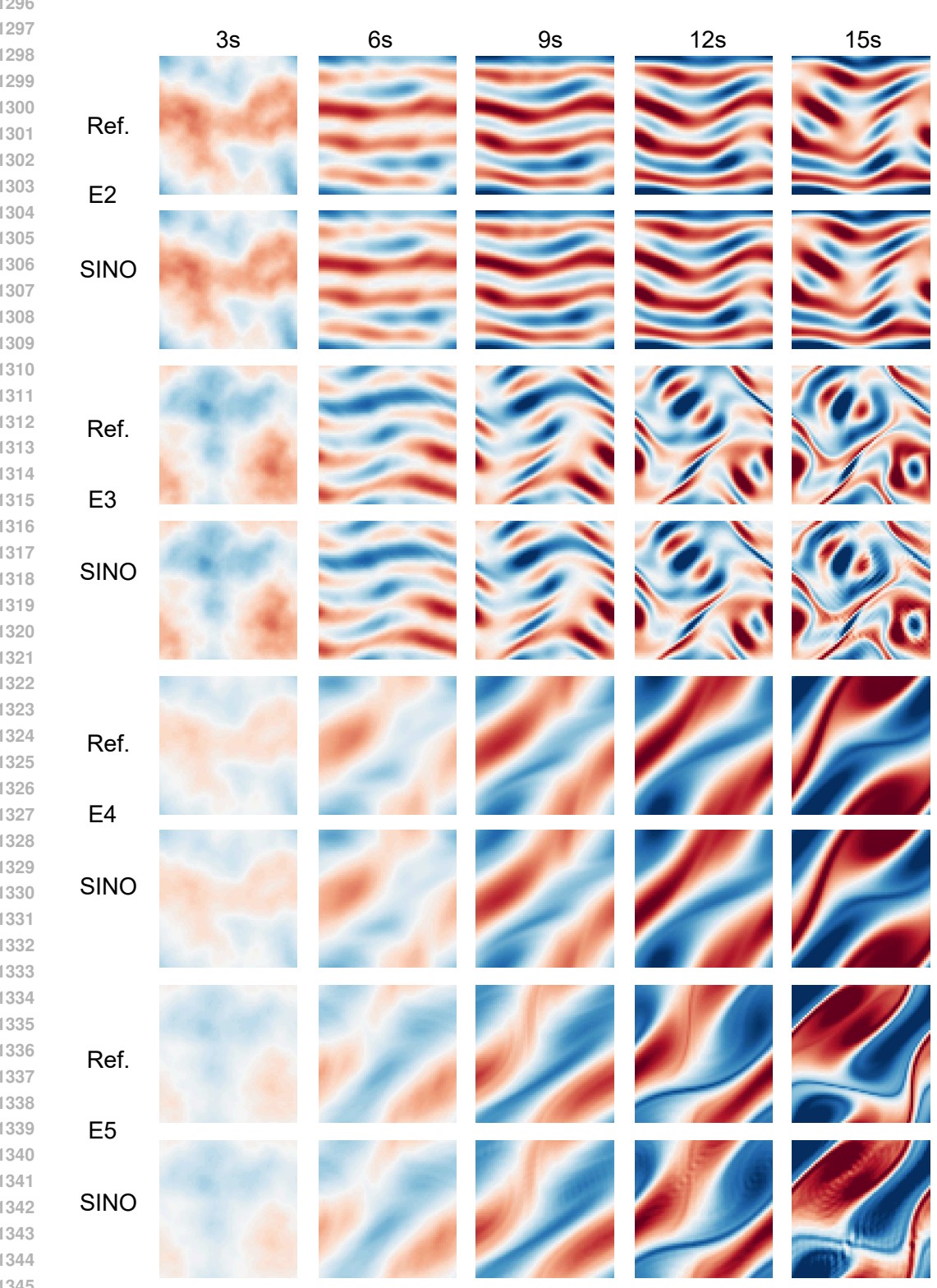

Figure 12: **Predicted trajectory of SINO.** Comparison of the vorticity field prediction trajectories for different scenarios (E2 to E5). The columns represent different prediction time steps at 3 s, 6 s, 9 s, 12 s, and 15 s.

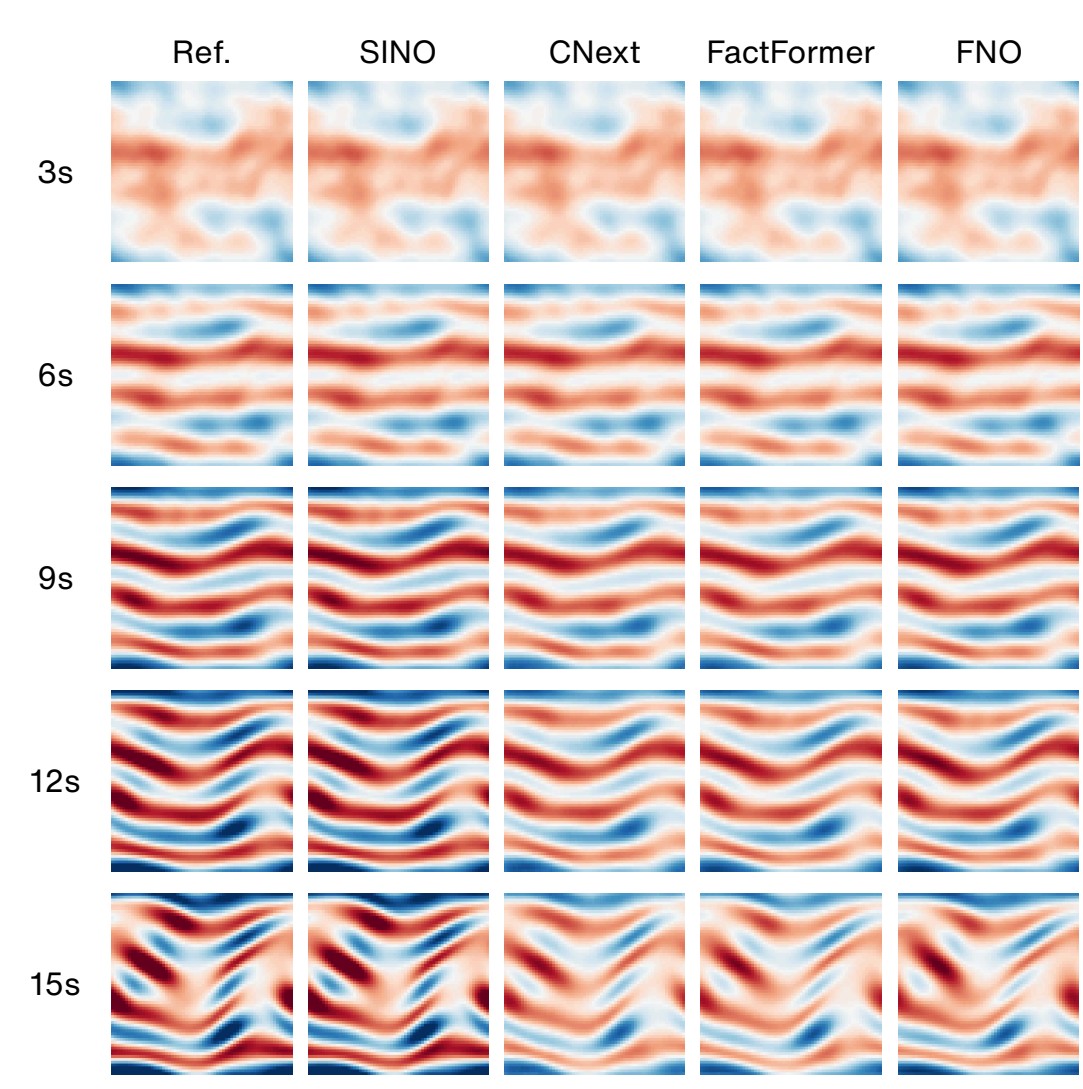

Figure 13: **In-distribution trajectories on IC0.** Comparison between SINO trained with only **5** trajectories and data-driven baselines trained with **1000** trajectories.

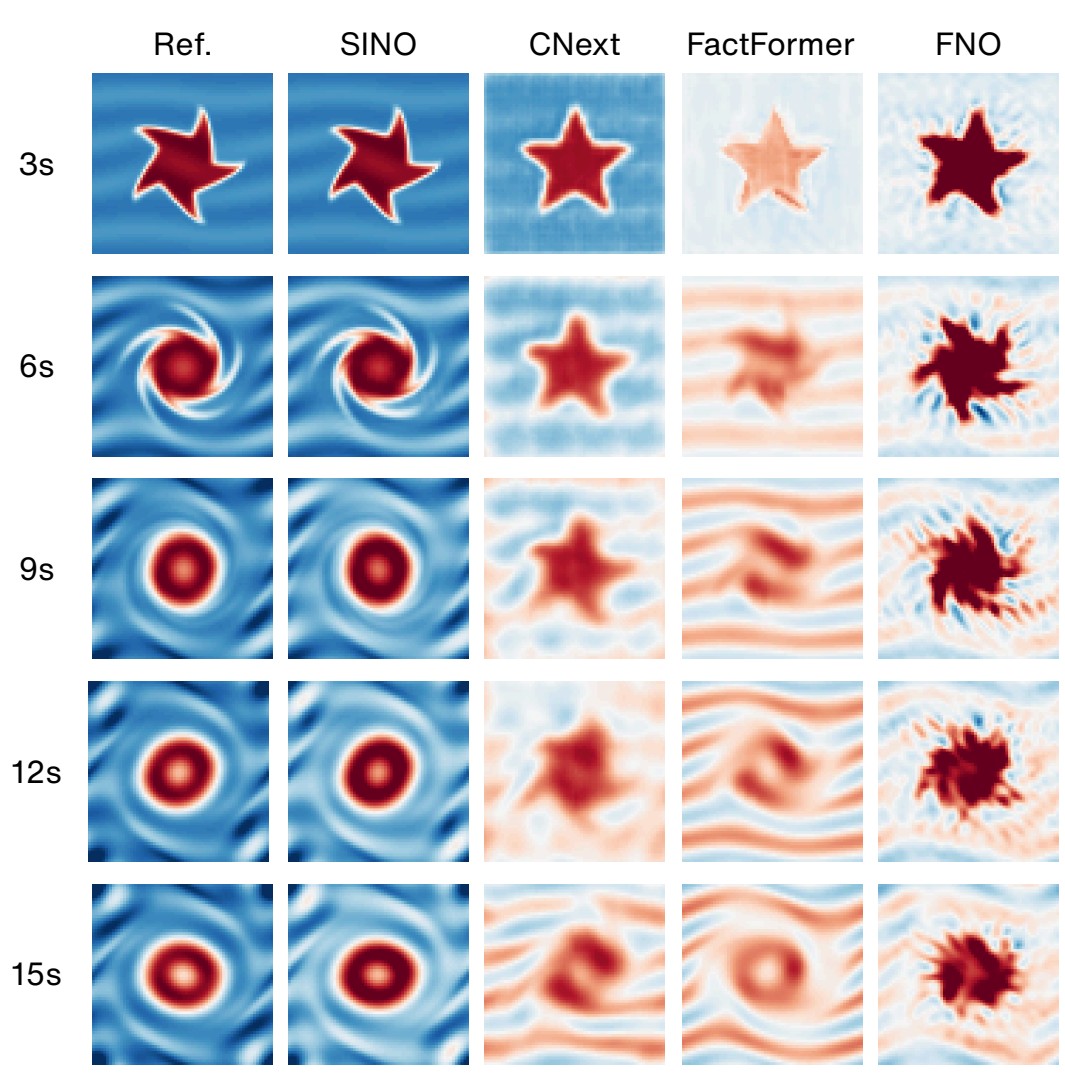

Figure 14: **OOD generalization trajectories on IC1 (star).** Comparison between SINO trained with only **5** trajectories and data-driven baselines trained with **1000** trajectories.

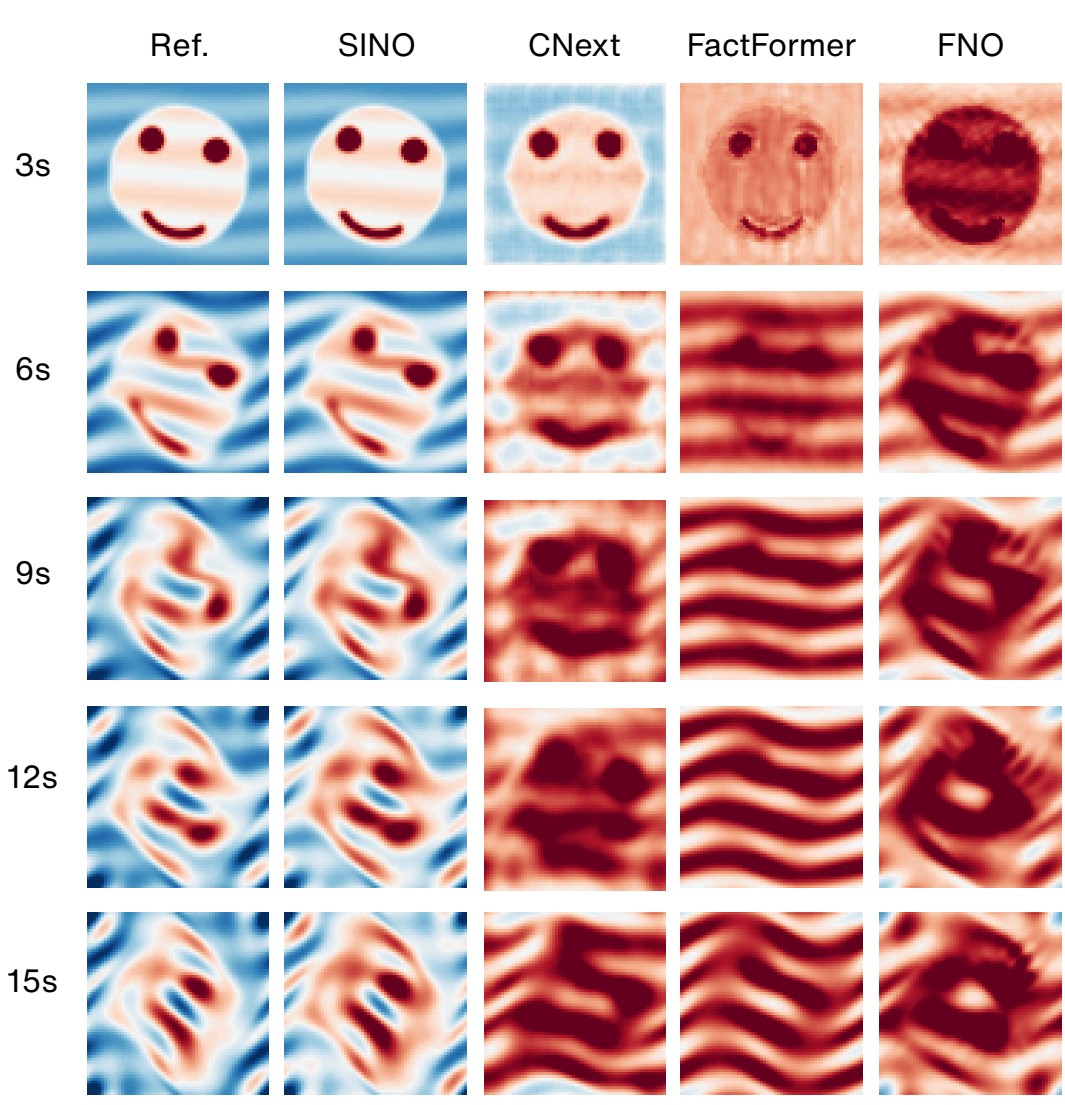

Figure 15: **OOD generalization trajectories on IC2 (smiley face).** Comparison between SINO trained with only **5** trajectories and data-driven baselines trained with **1000** trajectories.

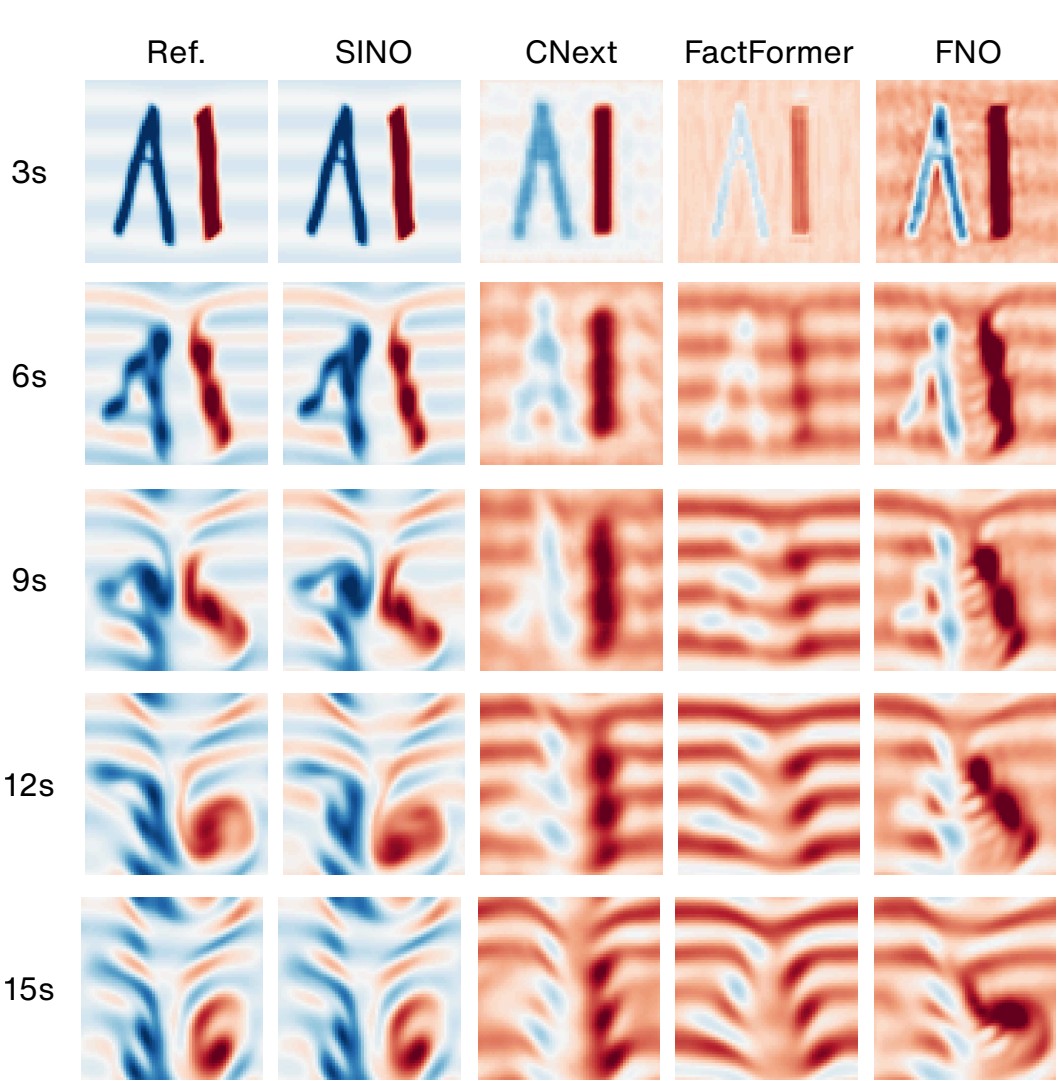

Figure 16: **OOD generalization trajectories on IC3 (the pattern 'AI').** Comparison between SINO trained with only **5** trajectories and data-driven baselines trained with **1000** trajectories.

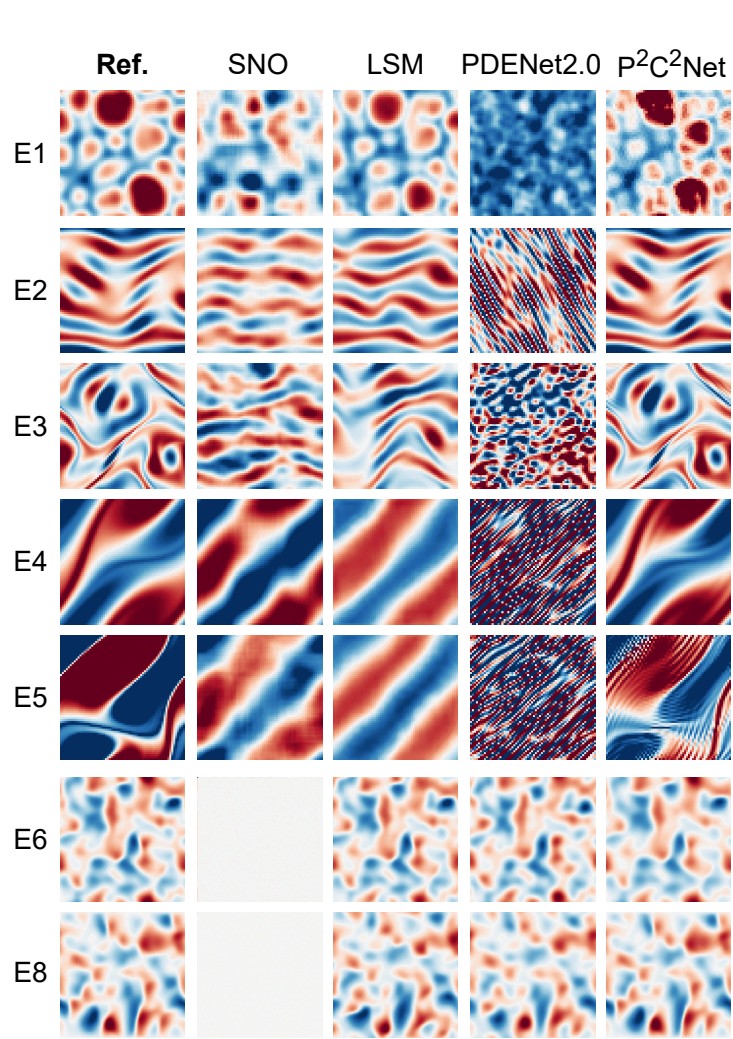

Figure 17: Forcasting results of SNO, LSM, PDENet2,0 and P$^2$C$^2$Net.

