# OpenReview forum: "Spectral-inspired Operator Learning with Limited Data and Unknown Physics"
_ICLR.cc/2026/Conference — Submitted to ICLR 2026_

### Official Review · Reviewer_kqv5 · 2025-10-16

**Soundness:** 3
**Presentation:** 3
**Contribution:** 3
**Rating:** 6
**Confidence:** 3

**Summary:**

This paper proposes a novel neural operator framework that learns to predict spatial derivatives and the combined right-hand side of partial differential equations (PDEs), which are then advanced in time using the RK4 scheme. The proposed method demonstrates superior data efficiency and predictive accuracy compared to baseline approaches on the 2D Kuramoto–Sivashinsky equation (KSE), 2D Navier–Stokes equations (NSE), and both 2D and 3D Burgers’ equations.

**Strengths:**

1. The paper is well structured and clearly presented. In particular, the details of SINO and its connection to spectral methods are thoroughly explained, accompanied by solid theoretical insights that enhance the overall understanding of the approach.
2. The effectiveness of SINO is comprehensively validated across a diverse set of benchmarks, including both 2D and 3D PDEs. Moreover, SINO exhibits strong robustness in low-data regimes and maintains high performance under out-of-distribution settings.

**Weaknesses:**

1. **Insufficient comparisons with baselines**: The integration of spectral methods with neural operators has been investigated in prior works such as SNO[1], LSM[2] and NMS[3]. A more comprehensive discussion of how spectral components are incorporated into model architectures and training strategies in these studies would further enrich the Related Works section. While NMS adopts a physics-informed training paradigm and may not be directly comparable, SNO and LSM share a more similar experimental setting and therefore serves as a suitable baseline for empirical evaluation.
2. **Experiments on more challenge tasks**: The authors state that the proposed method addresses challenges related to limited data and unknown physical models, which are common in scientific computing. However, concrete examples of such scenarios are not provided. I suggest that the authors include realistic examples of these scenarios to better illustrate the motivation and to evaluate the method’s effectiveness in practical settings. For instance, the practical design tasks in [4] could serve as a useful reference for such considerations.
3. **Scalability with respect to data**: Although the paper primarily focuses on the low-data regime, it is also important to assess how the proposed method scales with increasing amounts of training data. An analysis of data scalability would provide a more complete understanding of the method’s performance characteristics.


**Reference**

[1] Fanaskov, Vladimir Sergeevich, and Ivan V. Oseledets. "Spectral neural operators." Doklady Mathematics, 2023.

[2] Wu, Haixu, et al. "Solving High-Dimensional PDEs with Latent Spectral Models." ICML, 2023.

[3] Du, Yiheng, Nithin Chalapathi, and Aditi S. Krishnapriyan. "Neural Spectral Methods: Self-supervised learning in the spectral domain." ICLR 2024.

[4] Wu, Haixu, et al. "Transolver: A Fast Transformer Solver for PDEs on General Geometries." ICML 2024.

**Questions:**

Please see Weaknesses.

---

> ### Author Response · Authors · 2025-11-21
> **Response to Reviewer kqv5**
>
> Thank you for your time and valuable feedback, as well as your positive comments and interest in our paper.
>
>
> **Weakness 1: Insufficient comparisons with baselines.**
>
> **Reply:** Valuable suggestion! We have added **Section 3.4** to include a comprehensive discussion on how SNO and LSM incorporate spectral components as follows, and we have already discussed the NSM in the related work part.
>
> > **SINO differs from existing spectral methods in both *what* it learns in the frequency domain and *how* it introduces nonlinearity**. In the frequency domain, FNO uses a large spectral kernel to directly mix coefficients; SNO, AFNO, and DPOT apply MLPs directly to frequency-domain feature maps, transforming the learned coefficients with a black-box mapping; LSM projects the dynamics into a compact latent space and performs its basis operations. While expressive, these designs largely behave as black boxes in the spectral domain and offer limited physical meaning, which makes them struggle in few-shot settings. In contrast, SINO learns embeddings over frequency indices $k$ with Freq2Vec, encoding the prior that derivative multipliers are functions of $k$. Coupled with a simple multiplicative $\Pi$-block, SINO's intermediate features align well with the ground-truth physical operators such as $\partial_x \omega$ and $\Delta \omega$, thereby yielding strong inductive biases for learning the underlying physics.
>
>
>
> Following your advice, we have added **SNO** and **LSM** as baselines for empirical evaluation on benchmarks E1–E6. The comparative results are reported in **Table A** below and in the revised paper (**Section 5.2**).
>
> **Table A: Comparison with spectral-based baselines (Relative $\ell_2$ Error)**
>
> | Model | E1 | E2 | E3 | E4 | E5 | E6 |
> | :--- | :---: | :---: | :---: | :---: | :---: | :---: |
> | SNO | 0.6776 | 0.7774 | 0.8446 | 0.3526 | 0.4009 | 1.0001 |
> | LSM | 0.2652 | 0.6845 | 0.7293 | 0.3470 | 0.3664 | 0.1018 |
> | **SINO (ours)** | **0.0122** | **0.0110** | **0.0303** | **0.0031** | **0.0171** | **0.0008** |
>
> As shown in **Table A**, SINO consistently outperforms SNO and LSM across all datasets.
>
> **Weakness 2: Lack of realistic examples beyond canonical PDE benchmarks.**
>
> **Reply:** We thank the reviewer for this suggestion. We focus on settings with only a small number of high-fidelity simulations and no analytic PDE (common in wall-bounded/channel flows), and therefore include a more challenging mixed-boundary benchmark **E8 (MixedBC)** with Dirichlet BCs on the top/bottom and periodic BCs on the left/right. We also note that spectral methods are known to be less straightforward on irregular geometries; they typically require suitable coordinate mappings or carefully designed curvilinear grids to preserve accuracy and stability. Extending SINO beyond rectangular domains is therefore an important future direction for us. Furthermore, we need to highlight that, on regular domains, our contribution is significant: **SINO demonstrates that globally coupled dynamics (e.g., NSE) can be learned from limited trajectories without any PDE terms.** To our knowledge, it has not been convincingly addressed by any other prior operator-learning methods.
>
>
> **Table B: Relative $\ell_2$ error on E8 (MixedBC)**
>
> | Model   | Relative $\ell_2$ Error (E8, MixedBC) |
> | :- | :-: |
> | PeRCNN  | 0.0170   |
> | PeSANet  | 0.0967   |
> | CROP  | 0.9833  |
> | CNext | 0.2007 |
> | FactFormer  | 0.5908   |
> | FFNO  | 0.2837  |
> | FNO  | 0.4495  |
> | SNO  | 1.0001  |
> | LSM   | 0.0962 |
> | PDENet 2.0  | 0.2210  |
> |  P$^2$C$^2$Net (Solver Inside) | 0.0138    |
> | **SINO (ours)** | **0.0002**   |
>
> Extending SINO to non-periodic boundaries is natural: we simply replace the FFT with an appropriate spectral transform (e.g., Discrete Sine Transform (DST)) to handle the non-periodic components. The detailed settings and results are included in the revised paper (**Section 5.2**).
>
>
>
> **Weakness 3: Scalability with respect to data.**
>
> **Reply:** Thank you for your comments! We agree that data scalability is important. In **Appendix C.1 (Data Efficiency)** on KSE (E1), we vary the number of training trajectories from 1 to 16. As shown below, the relative $\ell_2$ error decreases as more data are provided and then saturates, indicating that SINO both benefits from additional data and is already effective in the low-data regime:
>
>  **Table C: Relative $\ell_2$ error of SINO with varying training dataset sizes on KSE (E1)**
>
> |Number of Trajectories|Relative $\ell_2$ Error|
> |:-|:-:|
> |1|0.0430|
> |2|0.0285|
> |4|0.0175|
> |8|0.0122|
> |16|0.0139|

---

> > ### Comment · Reviewer_kqv5 · 2025-11-27
> >
> > I thank the authors for their detailed and clear responses. Most of my concerns have been addressed and my score remains unchanged.

---

> > > ### Author Response · Authors · 2025-11-27
> > >
> > > Thank you for your support and reply.
> > >
> > > Best,
> > >
> > > The Authors

---

### Official Review · Reviewer_ZG9x · 2025-10-27

**Soundness:** 1
**Presentation:** 1
**Contribution:** 2
**Rating:** 2
**Confidence:** 4

**Summary:**

The paper introduces the Spectral-Inspired Neural Operator (SINO), a novel architecture designed to learn PDE dynamics from extremely limited data (as few as 2-5 trajectories) without prior knowledge of the governing equations. Inspired by classical spectral methods, SINO uses modules to learn spectral multipliers for derivatives and nonlinear interactions (with de-aliasing). SINO is evaluated on 2D and 3D PDE benchmarks, outperforming several data-driven & physics-encoded baselines, and demonstrating robust out-of-distribution generalization.

**Strengths:**

- The OOD generalization and super-resolution experiments demonstrate that SINO is able to approximate the underlying operator using only a very limited number of trajectories.
- The architecture is motivated by first principles, resulting in a learnable (physics-agnostic) version of classical spectral solvers and providing a good inductive bias for low-data regimes.
- The paper provides ablation studies confirming the necessity of each key component.

**Weaknesses:**

There are two main concerns:

1. In almost all experiments, the parameter count of the baselines is significantly larger than the one of SINO. Together with the relatively small grid of hyperparameters used for each method, it seems that all these methods are overfitting to the limited amount of training data. For a fair comparison, each baseline should be evaluated with several configurations that lead to a comparable parameter count. Moreover, it is unclear how much samples from each trajectory are taken for each method, in particular, since Table 3 shows different time-step sizes for DOL & POL methods.
2. The proposed model is not a surrogate model for accelerating the numerical solution of PDEs. It incurs a similar inference time as a solver since it still relies on a RK4 integrator with sufficiently small steps. In this sense, there is no point in comparing to data-driven surrogate models. The focus of the paper should be on learning/discovering the underlying physical operator from limited data and comparing to other baselines in that domain. While the paper propose *operator distillation* as a workaround, this only shows that the underlying operator can be sufficiently well approximated by SINO such that it can act as a synthetic data generator.

Other concerns:

- Apart from the RK scheme used for time-stepping, the design of SINO is very similar to variants of FNOs, in particular https://arxiv.org/abs/2403.12553 for nonlinear mixing in the channel dimension and https://arxiv.org/pdf/2111.13587, https://arxiv.org/pdf/2403.03542 for MLPs in the Fourier space.
- In its current form, the method is dependent on regular grids and periodic boundary conditions, excluding a vast number of real-world physics and engineering problems.
- It would be good to evaluate other metrics such as spectra.
- As mentioned above, only a few hyperparameter settings are tested for each baseline. In particular, this only includes a single training strategy (pushforward trick) with two learning rates.
- The universality result is relatively standard and it seems to also follow from https://arxiv.org/abs/2304.13221
- It is unclear how the low-pass filter is adapted in the super-resolution results.

**Questions:**

See Weaknesses above

---

> ### Author Response · Authors · 2025-11-21
> **Response to Reviewer ZG9x [1/3]**
>
> Thank you for your time and valuable feedback.
>
> **Weakness 1: Unfair comparison regarding parameter counts.**
>
> **Reply:** Thank you for raising this concern.
>
> **1. On the fairness of parameter-count matching.** We do not enforce strict parameter parity across different architectures because **SINO and other architectures have inherently different capacity–complexity trade-offs**. For instance:
>
> (i). **SINO**’s parameterization scales as $\mathcal{O}(C^2)$ plus a lightweight Freq2Vec MLP;
>
> (ii) **FNO** scales as $\mathcal{O}(C^2 K^2)$, where $C$ is the channel width and $K$ is the number of retained Fourier modes.
>
> To match SINO’s parameter count, one would have to simultaneously shrink FNO’s $C$ and $K$ to very small values (typically $2-6$), which severely impairs its ability to learn high-frequency signals and limits the model’s expressivity. The same reasoning applies to CNext or other data-driven baselines. To verify this, we conducted extensive ablation studies on CNext and FNO by varying their parameter counts from approximately $\mathcal{O}(10^3)$ to $\mathcal{O}(10^5)$, covering configurations that are **smaller than, comparable to, and larger than** SINO. The results are summarized in **Table A** and **Table B** below.
>
> As shown in Tables A and B, reducing the parameter counts of the data-driven baselines **does not** improve their performance. **The reported results in our paper significantly outperformed these lightweight models.**
>
>
> **Table A: CNext performance with varying parameter counts (relative $\ell_2$ error)**
>
> |  Model (\#Params.)   | Layer | Width | E2     | E3     | E4     | E5     |
> | :-  | :---: | :---: | :---:  | :---:  | :---:  | :---:  |
> | **CNext (Original)** | 2~4 | 16~64 | **0.7060** | **0.7388** | **0.2877** | **0.3153** |
> | CNext (1,893) | 1 | 2| 0.8989 | 0.9993 | 0.5512 | 0.7847 |
> | CNext (5,065) | 1 | 4 | 1.0282 | 0.9797 | 0.3851 | 0.4193 |
> | CNext (6,125) | 2 | 2 | 0.8005 | 0.9570 | 0.4051 | 0.4137 |
> | CNext (15,249) | 1 | 8 | 0.8757 | 0.9557 | 0.3903 | 0.4206 |
> | CNext (18,201) | 2 | 4 | 0.7592 | 0.8362 | 0.3297 | 0.3296 |
> | CNext (60,209) | 2 | 8 | 0.7502 | 0.8632 | 0.3158 | 0.3385 |
>
>
> **Table B: FNO performance with varying parameter counts (relative $\ell_2$ error)**
>
> | Model (\#Params.) | Modes | Width | E2 | E3 | E4 | E5 |
> | :---           | :---: | :---: | :---: | :---: | :---: | :---: |
> | **FNO (Original)** | 8~16 | 12~64 | **0.5210** | **0.6253** | **0.2282** | **0.2574** |
> | FNO (1,889)   | 2 | 4  | 0.8008 | 0.8420 | 0.3278 | 0.3622 |
> | FNO (4,961)   | 4 | 4  | 1.0253 | 0.8635 | 0.5141 | 0.5030 |
> | FNO (5,697)   | 2 | 8  | 0.9005 | 0.8672 | 0.2578 | 0.2641 |
> | FNO (10,081)  | 4 | 6  | 0.7786 | 0.9726 | 0.2771 | 0.3189 |
> | FNO (11,681)  | 2 | 12 | 0.8531 | 0.8797 | 0.3713 | 0.4370 |
> | FNO (17,249)  | 8 | 4  | 0.9027 | 1.0780 | 0.4703 | 0.4922 |
> | FNO (38,465)  | 6 | 8  | 0.6661 | 0.8215 | 0.2492 | 0.2758 |
> | FNO (67,137)  | 8 | 8  | 0.5292 | 0.6664 | 0.2699 | 0.3096 |
> | FNO (85,409)  | 6 | 12 | 0.5735 | 0.7368 | 0.2536 | 0.2794 |
>
> **2. Data and sampling consistency (DOL vs. POL).** Thank you for your comments. Although the time-step sizes for DOL and POL are different, every frame in each training trajectory has an equal probability of being sampled during training. In other words, both DOL- and POL-based models are exposed to the **same number of frames per trajectory** throughout training.

---

> ### Author Response · Authors · 2025-11-21
> **Response to Reviewer ZG9x [2/3]**
>
> **Weakness 2: Compared to other physical operators**
>
> **Reply:** Thank you for your valuable suggestions. SINO tackles a challenging task: predicting **globally coupled systems (such as NSEs) with unknown physics and limited data**. To the best of our knowledge, **all existing approaches fail to address this challenging setting:**
>
> **(i) Purely data-driven neural operators** typically require a large amount of training data.
>
> **(ii) Physics discovery methods** are often confined to systems governed by local operators (e.g., Burgers' equation or reaction-diffusion systems) and struggle with globally coupled systems like the NSE without additional priors. In our revised version, we have tested three discovery methods, including **PeRCNN**, **PESANet**, and the newly added **PDENet2.0**, as shown in **Table C**. Furthermore, other discovery methods cannot simulate unseen physical fields, and few of them can successfully discover or model NSE dynamics.
>
> **(iii) Hybrid methods** rely on known physics. We compared SINO against a new baseline named **P$^2$C$^2$Net** in **Table C**, which **needs full knowledge of the governing equations** and embeds a solver within the network. Even against the favorable setting of P$^2$C$^2$Net, SINO achieves superior overall accuracy across most benchmarks.
>
> Therefore, SINO accomplishes a task that current state-of-the-art methods cannot. **It represents a distinct methodology for handling global coupling without physical priors, rather than a mere incremental performance improvement.** We believe this contribution offers a significant step forward for the field.
>
>
> **Table C: New baselines (relative $\ell_2$ error).  Note that P$^2$C$^2$Net requires full physical information and includes a classical numerical solver inside.**
>
> | Model | E1| E2| E3| E4| E5| E6|
> |--|-|-|--|-|-|-|
> | PDENet 2.0                | 0.4703   | 0.9587   | 0.9737   | 0.9027   | 1.1070   | 0.2254           |
> | P$^2$C$^2$Net  | 0.3192   | 0.0346   | **0.0215**   | 0.0526   | 0.1015   | 0.0110           |
> | **SINO (ours)**           | **0.0122** | **0.0110** | 0.0303 | **0.0031** | **0.0171** | **0.0008** |
>
>
>
> **Weakness3: Relation to FNO Variants (CoDA-NO, AFNO, DPOT)**
>
> **Reply:** Thank you for your comments. While SINO shares a broad spectral–operator perspective with recent FNO variants, its design differs **fundamentally and conceptually** from these works.
>
> (i) **CoDA-NO**: The nonlinear *mixing of channels* in CoDA-NO happens in the **spatial domain** through an attention mechanism, which lacks clear physical interpretability. In contrast, **SINO does not rely on any attention operation** to mimic nonlinearity. Instead, it introduces the **Π-Block**, which explicitly models the nonlinear term of the underlying PDE through element-wise multiplicative interactions. This simple structure preserves **physical interpretability** while maintaining efficiency.
>
> (ii) **AFNO and DPOT**: The roles of MLPs are completely different. In AFNO and DPOT, the MLP operates on the **frequency-domain feature maps** $\hat{X}$, i.e., it mixes or transforms the learned spectral coefficients directly. In **SINO**, however, the MLP acts on the **frequency indices $k$** to simulate derivative multipliers rather than on $\hat{X}$ itself.
>
> Therefore, **SINO introduces a fundamentally different form of nonlinearity and frequency representation**, resulting in a model that is both **interpretable and physically grounded** rather than merely an architectural variant of FNO. We have added **Section 3.4** to discuss the difference between SINO and other neural spectral methods.
>
> **Weakness 4: Applicability to non-periodic boundary conditions.**
>
> **Reply:** Insightful question! To address this, we have introduced a more challenging dataset (**E8**) involving the 2D Burgers' equation with **mixed boundary conditions** (Dirichlet BCs on top/bottom and Periodic BCs on left/right). Extending SINO to non-periodic boundaries is natural: we simply replace the FFT with an appropriate spectral transform (e.g., Discrete Sine Transform (DST)) to handle the non-periodic components. The detailed settings and results are included in the revised paper (**Table 1, Section 5.2**).
>
> **Table D: Relative $\ell_2$ error on E8 (MixedBC)**
>
> | Model | Relative $\ell_2$ Error (E8, MixedBC) |
> | :--- | :---: |
> | PeRCNN | 0.0170 |
> | PeSANet | 0.0967 |
> | CROP | 0.9833 |
> | CNext | 0.2007 |
> | FactFormer | 0.5908 |
> | FFNO | 0.2837 |
> | FNO | 0.4495 |
> | SNO | 1.0001 |
> | LSM | 0.0962 |
> | PDENet 2.0 | 0.2210 |
> | P$^2$C$^2$Net | 0.0138 |
> | **SINO (ours)** | **0.0002** |
>
> As shown in **Table D**, SINO achieves a relative $\ell_2$ error of **0.0002**, significantly outperforming the baseline methods. This demonstrates SINO's flexibility and superior performance in handling non-periodic boundary conditions.

---

> ### Author Response · Authors · 2025-11-21
> **Response to Reviewer ZG9x [3/3]**
>
> **Weakness 4: Spectral evaluation.**
>
> **Reply:** Thank you for your insightful suggestions! We have added energy spectra plots for the NSE experiments in **Appendix C.6 (Fig. 10)**. The spectral energy distribution provides insight into how each model captures multi-scale dynamics and energy transfer mechanisms across wavenumbers. In all cases, SINO maintains close alignment with the DNS spectrum across the full range of $\kappa$, effectively preserving both large-scale and fine-scale structures.
>
> **Weakness 5: Few hyperparameter settings.**
>
> **Reply:** **Thank you for raising this point.** In fact, our study involved a **comprehensive hyperparameter exploration** across a large and diverse experimental setup. Specifically, we evaluated 12 different operator-learning methods on 8 datasets, and for **each method–dataset pair**, we tested **10–20 distinct key hyperparameter configurations** (including learning rate, network width/depth, and spectral modes). In total, we trained **over 1,000 neural operators**. All models were trained **until convergence**, and the **training loss curves** (provided in Appendix C.5) clearly show that each baseline reached a stable regime.
>
> Given this scale of experiments, the overall tuning effort in our work **exceeds that of several well-known PDE benchmark papers**, such as *The Well* [1] and *PDEBench* [2]. These benchmarks typically tune hyperparameters on **only one representative case** and then apply the same configuration across all cases. However, as shown in **Table 4 (Appendix B.2)**, the optimal architectures and hyperparameters in our experiments vary substantially across different cases.
>
> Furthermore, once a model is trained to convergence, **fine-tuning the learning rate or number of iterations generally has negligible impact** on performance. Therefore, we believe our tuning protocol is both **appropriate and more rigorous** than those adopted in most existing PDE benchmark studies.
>
> **Weakness 6: The universality result.**
>
> **Reply:** Thank you for the pointer. While we appreciate the connection, our theorem has a significant difference with the mentioned results, which addresses a **different object under different structural assumptions**:
>
> 1) **No lifting in SINO.** The universality in this paper is proved for architectures that explicitly **lift** the input function to a higher-dimensional latent field via an encoder, apply a nonlocal/nonlinear hidden layer, and then **project** back. This lifting–projection scaffold is **crucial** to their result. **SINO does not perform any lifting/projection**: it operates at the original channel dimension, using frequency-index embeddings (Freq2Vec) and a physically interpretable $\Pi$-block that mirrors PDE nonlinearities.
>
> 2) **Different approximation target.** The paper proves universality for **continuous operators between function spaces** (e.g., $C^s \to C^{s'}$). By contrast, our theorem concerns the **approximation of PDE differential operators**, i.e., we target the derivative operator underlying the evolution, not an arbitrary continuous map.
>
> Therefore, the cited result is complementary but does **not** subsume ours: it pertains to a different architectural regime (with lifting and explicit nonlocal mixing) and a different universality notion.
>
> **Weakness 7:Unclear adaptation of the low-pass filter in super-resolution experiments.**
>
> **Reply:** Thank you for your question. The low-pass filter in super-resolution experiments keeps the 2/3 setting.
>
> [1] Ohana R, McCabe M, Meyer L, et al. The well: a large-scale collection of diverse physics simulations for machine learning[J]. Advances in Neural Information Processing Systems, 2024, 37: 44989-45037.
>
> [2] Takamoto M, Praditia T, Leiteritz R, et al. Pdebench: An extensive benchmark for scientific machine learning[J]. Advances in Neural Information Processing Systems, 2022, 35: 1596-1611.

---

### Official Review · Reviewer_vLmH · 2025-10-29

**Soundness:** 2
**Presentation:** 2
**Contribution:** 2
**Rating:** 2
**Confidence:** 4

**Summary:**

This paper proposes SINO (Spectral-Inspired Neural Operator), a neural architecture designed to learn PDE dynamics from a very small number of trajectories without explicit knowledge of governing equations. The model mimics classical spectral solvers by operating in the frequency domain through a learnable “Freq2Vec” module, modeling nonlinear terms via multiplicative Π-blocks, and enforcing numerical stability using the 2/3 de-aliasing rule and RK4 time integration. Experiments on periodic PDEs (KSE, NSE, Burgers 2D/3D) claim state-of-the-art accuracy, outperforming existing neural and physics-encoded operators by 1–2 orders of magnitude.

**Strengths:**

- The authors propose an interesting surrogate architecture that directly mimics spectral numerical solvers. The inclusion of classical ingredients such as the 2/3 de-aliasing rule and Runge–Kutta-4 integration demonstrates a clear attempt to build numerical stability into the rollout process, something that many prior neural PDE solvers neglect.

- By structuring the model around Fourier transforms and multiplicative nonlinear interactions, SINO incorporates strong physics-motivated priors (e.g., global coupling in the frequency domain, multiplicative nonlinearities akin to convection terms). These choices may help mitigate overfitting in low-data settings.

- Conceptually, the paper highlights an appealing direction: combining spectral-method heuristics with learnable neural representations. Even if the implementation details are debatable, the underlying idea of treating neural operators as extensions of traditional numerical solvers is valuable for future research.

**Weaknesses:**

- Clarity and Notation

The paper is poorly written and the notation is vague, making it difficult to follow the technical exposition.
Section 3.3 (Architecture components), in particular, is imprecise and lacks formal definitions for core modules.

- Architectural Novelty Is Overstated

The “Spectral Learning Block” is highly similar to the Fourier layer in the Fourier Neural Operator (FNO). Both perform Fourier transforms, apply learnable frequency-domain multipliers, and inverse-transform the results. In FNO, this is realized via a learned matrix  R(k) operating directly on spectral coefficients. In SINO, this role is played by the Freq2Vec MLP that maps frequency indices to multipliers  Hence, the claimed novelty of “automatically capturing spatial derivatives via frequency embeddings” largely duplicates FNO’s existing design, merely reparameterized via an MLP. This similarity is never acknowledged or discussed, which undermines the claimed originality.

- Experimental Fairness and Reproducibility

(a) No released code

The supplementary material lacks any code or configuration files, making it impossible to verify the experimental claims.
Given the unusually large reported performance gap (1–2 orders of magnitude), this lack of transparency is problematic.

(b) Likely unfair training setup

The baselines are trained under a setup that gives SINO a massive structural advantage. SINO is trained and integrated with a fine time step (Δt ≈ 1e−3–1e−2) and Runge–Kutta 4 (RK4) integration, while the data-driven baselines (FNO, FFNO, CNext, etc.) are trained with much coarser Δt (0.2–1) and single-step rollouts. This means that SINO performs 100–1000× more effective temporal updates per trajectory, producing thousands of additional training samples and much higher temporal resolution. Given only 2–5 trajectories in total, the data-driven baselines effectively see only a few unique training examples and rapidly overfit. Thus, the baselines are not trained in a setting consistent with their intended design. This leads to an unfair comparison that strongly favors SINO’s numerically stable formulation.

(c) Baseline training likely under-tuned

Although the paper mentions a small “grid search,” the sweeps are extremely limited (few values for channel count or Fourier modes only).
No optimization over learning rates, batch sizes, regularization, normalization, or data augmentation is reported.
In such a low-data regime, these hyperparameters are crucial; omitting them almost guarantees poor performance for FNO-type models.
The “NaN” and “>1” errors in Table 1 likely reflect training instability or overfitting, not inherent architectural limitations.

- Overly Smooth and Favorable Benchmarks

All benchmark PDEs (KSE, NSE, Burgers) are periodic, smooth, and globally coupled, which are exactly the scenarios in which spectral methods excel. Since SINO is explicitly designed as a spectral-inspired model with periodic boundary assumptions, these datasets do not test its robustness. No non-periodic, multi-scale, or irregular-domain tasks are included. The authors should have evaluated SINO on more challenging benchmarks such as PDE-Gym [1] or PDEBench [2], which include non-periodic or discontinuous dynamics.

- Inference Efficiency Defeats the Point of Neural Operators

The inference time of SINO is comparable to that of a traditional spectral solver (Figure 8). This undermines the core purpose of neural operators, whose value lies in being orders of magnitude faster than classical solvers at the cost of some accuracy. SINO is effectively a slow, learnable surrogate of a spectral solver, not a practically deployable fast operator. While the authors introduce a “distilled” SINO-FNO variant for speed, its training depends on the expensive SINO teacher, defeating the point of efficiency.

- Theoretical Analysis Is Hand-Wavy

The so-called “universal approximation theorem” (Theorem 1) is not rigorous. The argument establishes approximation only for a fixed input function u,  not uniformly over admissible functions. The constants C_j​ (lines 762-765) depend on u, so no operator-norm or uniform bound is shown. Hence, this is not a true universal approximation result for operators.

- Lack of Conceptual Rigor and Reproducibility

The description of training dynamics (warm-up steps, rollouts) is inconsistent across models. It is unclear whether identical normalization, boundary padding, or Fourier transform conventions were used for all methods. Reported “NaN” results for multiple baselines suggest missing stability controls or inconsistent preprocessing. Without the released code, these ambiguities make the empirical findings non-reproducible and potentially unreliable.

____

[1] Herde, M., Raonic, B., Rohner, T., Käppeli, R., Molinaro, R., de Bézenac, E., & Mishra, S. (2024). Poseidon: Efficient foundation models for pdes. Advances in Neural Information Processing Systems, 37, 72525-72624.

[2] Takamoto, M., Praditia, T., Leiteritz, R., MacKinlay, D., Alesiani, F., Pflüger, D., & Niepert, M. (2022). Pdebench: An extensive benchmark for scientific machine learning. Advances in Neural Information Processing Systems, 35, 1596-1611.

**Questions:**

How is the “Spectral Learning Block” fundamentally different from the Fourier layer in FNO, which also learns frequency-domain multipliers?

- Were all models trained with the same temporal discretization and integration scheme, or does SINO’s RK4 setup give it an implicit advantage in rollout stability?

- How do the authors ensure that data-driven baselines are not underfitting or overfitting given the extremely small number of training trajectories?

- Since all benchmarks are periodic and spectral in nature, how would SINO behave on non-periodic or multi-scale PDEs where spectral assumptions break down?

- Theorem 1 seems to establish approximation only for a fixed function u. How does this guarantee generalization across functions in operator space?

The authors should also review the Weaknesses section for additional (implicit) questions and points raised.

**Details Of Ethics Concerns:**

/

---

> ### Author Response · Authors · 2025-11-21
> **Response to Reviewer vLmH [1/2]**
>
> Thank you for your time and valuable feedback.
>
> **Weakness 1: Clarity and Notation.**
>
> **Reply:** Thank you for your comments. To make our revision maximally helpful, could you please kindly indicate which symbols or passages were unclear? If you can point to specific lines or sentences, we will address them directly in the revised version.
>
> **Weakness 2 & Question 1: Architectural Novelty Is Overstated.**
>
> **Reply:** Thank you for your comments. However, we respectfully disagree with the statement. SINO is fundamentally distinct from FNO in three key aspects regarding efficiency, nonlinear mechanism, and interpretability:
>
> - **Parameter Efficiency in Frequency Domain:** FNO parameterizes the spectral kernel via a large, dense tensor scaling as $\mathcal{O}(C^2 K^2)$, where $C$ is the channel width and $K$ is the number of retained Fourier modes. In contrast, SINO employs Freq2Vec to learn the explicit functions of frequency coordinates, significantly reducing parameter complexity and improving generalization.
> - **Physical Meaning of Non-linearity:** FNO relies on an MLP in the spatial domain for channel mixing, a standard deep learning operation that lacks explicit physical semantics. SINO utilizes the $\Pi$-block to perform multiplicative interactions, which concisely simulate specific non-linear dynamics (e.g., convection terms) inherent in governing PDEs.
> - **Interpretability:** SINO offers superior physical consistency. Both the Freq2Vec and the $\Pi$-block are structurally designed to align with physical operators, making the model interpretable (as demonstrated in Fig. 4), whereas FNO remains a black-box approximator.
>
> We have added **Section 3.4** to include a comprehensive discussion on how SINO differs from other neural operators.
>
>
> **Weaknesses 3 and 7: Experimental Fairness and Reproducibility.**
>
> >(a): No released code.
>
> **Reply:** Thank you for your comments. **We have released code to the supplementary material.**
>
> >(b) & Question 2: Likely unfair training setup.
>
> **Reply:** We respectfully disagree with the claim that SINO receives “100–1000× more effective temporal updates” than the data-driven baselines. All methods, including SINO, FNO, FFNO, CNext, etc., are trained on **exactly the same set of spatio-temporal snapshots**: for each trajectory, we subsample a fixed number of frames at constant physical time intervals and use these as supervision. During training, SINO uses RK4 to advance from one observed frame to the next over the *same* physical time interval as the baselines; this does not create additional training examples, but only evaluates the learned right-hand-side operator multiple times within a single integration step. For fairness, we also use one-step (and short multi-step) rollouts on the *same time grid* for FNO/CNext, so they see the same number of unique frames per trajectory as SINO. We also remark that several POL baselines use RK4 integration with similar time discretizations; nevertheless, they do not achieve the same level of accuracy as SINO under the same low-data, unknown-physics setting.
>
> >\(c) & Question 3:Baseline training likely under-tuned
>
> **Reply:** Thank you for raising this point. In fact, our study involved a **comprehensive hyperparameter exploration** across a large and diverse experimental setup. Specifically, we evaluated 12 different operator-learning methods on 8 datasets, and for **each method–dataset pair**, we tested **10–20 distinct key hyperparameter configurations** (including learning rate, network width/depth, and spectral modes). In total, we trained **over 1,000 neural operators**. All models were trained **until convergence**, and the **training loss curves** (provided in Appendix C.5) clearly show that each baseline reached a stable regime.
>
> Given this scale of experiments, the overall tuning effort in our work **exceeds that of several well-known PDE benchmark papers**, such as *The Well* [1] and *PDEBench* [2]. These benchmarks typically tune hyperparameters on **only one representative case** and then apply the same configuration across all cases. However, as shown in **Table 4 (Appendix B.2)**, the optimal architectures and hyperparameters in our experiments vary substantially across different cases.
>
> Furthermore, once a model is trained to convergence, **fine-tuning the learning rate or number of iterations generally has negligible impact** on performance. Therefore, we believe our tuning protocol is both **appropriate and more rigorous** than those adopted in most existing PDE benchmark studies.

---

> ### Author Response · Authors · 2025-11-21
> **Response to Reviewer vLmH [2/2]**
>
> **Weakness 4 & Question 4: Overly smooth and favorable benchmarks.**
>
> **Reply:** Thank you for your comments. However, we respectfully disagree with this characterization because our OOD experiments in **Section 5.3** have already targeted **highly non-smooth** initial conditions, explicitly probing robustness out of the training distribution. Furthermore, benchmark suites such as PDE-Gym and PDEBench also center around equations like KSE, NSE, and Burgers; our choice of test problems is consistent with this established practice for evaluating neural PDE solvers. Finally, in the revision, we additionally include a mixed-boundary benchmark **E8 (MixedBC)** with Dirichlet BCs on the top/bottom and periodic BCs on the left/right.
>
>
> **Table A: Relative $\ell_2$ error on E8 (MixedBC)**
>
> | Model | Relative $\ell_2$ Error (E8, MixedBC) |
> | :--- | :---: |
> | PeRCNN | 0.0170 |
> | PeSANet | 0.0967 |
> | CROP | 0.9833 |
> | CNext | 0.2007 |
> | FactFormer | 0.5908 |
> | FFNO | 0.2837 |
> | FNO | 0.4495 |
> | SNO | 1.0001 |
> | LSM | 0.0962 |
> | PDENet 2.0 | 0.2210 |
> | P$^2$C$^2$Net | 0.0138 |
> | **SINO (ours)** | **0.0002** |
>
>
> **Weakness  5: Inference Efficiency Defeats the Point of Neural Operators**
>
>
> **Reply:** We respectfully disagree with this assessment. A full rollout of SINO is **faster than our reference spectral solver**, even while operating in the much harder regime of **unknown physics and very limited data**. In this setting, SINO is the most accurate and already a competitive surrogate to the numerical solver. Moreover, SINO is explicitly designed as an **operator learner**: it learns a resolution-agnostic approximation of the PDE right-hand side (via Freq2Vec and $\Pi$-blocks) that can be evaluated on different grids. This capability aligns precisely with the goal of neural operator learning, rather than mere “one-shot acceleration” on a fixed mesh.
>
> The role of SINO-FNO is complementary, not contradictory. Once SINO has learned a physically consistent operator from a handful of high-fidelity trajectories, it can generate abundant, cheap **synthetic rollouts** under **unknown governing equations**. These SINO-generated trajectories allow us to train fast students (such as FNO) that inherit both the speed of standard neural surrogates and the accuracy of SINO in the low-data regime.
>
> **Weakness 6 and Question 5: Theoretical Analysis Is Hand-Wavy.**
>
> **Reply:** Thank you for the careful comment. We have revised Theorem 1 to make the uniform nature of the approximation explicit.
>
> [1] Ohana R, McCabe M, Meyer L, et al. The well: a large-scale collection of diverse physics simulations for machine learning[J]. Advances in Neural Information Processing Systems, 2024, 37: 44989-45037.
>
> [2] Takamoto M, Praditia T, Leiteritz R, et al. Pdebench: An extensive benchmark for scientific machine learning[J]. Advances in Neural Information Processing Systems, 2022, 35: 1596-1611.

---

> > ### Comment · Reviewer_vLmH · 2025-11-26
> >
> > (1) As for the clarity of writing, I was referring to the whole Section 3: it is difficult to follow which layer is applied where. However, I do not consider this a major weakness, and I do not think it needs to be discussed further.
> >
> > (2) Regarding the architecture, I agree that the authors’ design is more compact than the FNO. I appreciate the addition of Section 3.4. I believe this significantly strengthens the paper.
> >
> > (3) Regarding the choice of experiments, while appreciating an additional experiment, I strongly disagree with the claim that they are challenging. For instance, the NSE case is a highly non-turbulent setting, which is why I suggested more demanding benchmarks in my initial review. Moreover, testing the model on the Burgers equation, as is common in our field, does not by itself demonstrate the model’s capabilities on genuinely difficult tasks.
> >
> > (4) I apologize for my earlier statement about having an N-times larger number of effective samples. This confusion arose because it is not clear what the POL and DOL grids are, and where each of them is used. According to your code, the default dt in the SINO model initialization, as well as in the test script, is 0.005. This suggests that the POL discretization is used internally in SINO. The SINO time-stepping
> >
> > $u_{t+1} = u_t + dt_{SINO} * (k1 + 2 * k2 + 2 * k3 + k4) / 6$
> >
> > induces very small and very stable residual updates (i.e. there is hard skip connection $u_t \to u_{t+1}$).
> >
> > Still, a fundamental question remains unanswered: why do all other tested models that many researchers have worked on fail, while yours is successful? This suggests that something might be unfair in the training or evaluation setup. For example, what happens if you train an FNO using your RK4 stepping or even simple Euler stepping, under the same regime? Does its performance improve significantly, or is there something intrinsically special in SINO architecture that makes it outperform everything else by a huge margin?
> >
> > Looking at Figure 9, it appears that all baselines enter an overfitting regime after only about 1k iterations. This suggests that the baselines are not particularly well-trained (even if multiple hyperparameter sets were explored).  Again, it would be fair to test baselines in a similar residual-learning setup, with an explicit hard skip connection
> >
> > $u_{t+1} = u_t + f(u_t)$
> >
> > RK4 time-stepping strategy acts as a strong stabilization trick for SINO. If no AR stabilization is used for the baselines, long rollouts during both training and inference may hurt their performance.
> >
> > The authors should clearly explain how all models are trained: which temporal regime is used, what is the effective dt for each model, whether training is AR or one-step, and whether the models learn the residual or the full forward operator (residual or non-residual training). It would also be important to report training times and inference rollout settings (including dt and horizon) for each model to ensure that there is no hidden advantage for SINO in terms of time stepping or rollout length during training/inference. Most of these important details are not in the paper.
> >
> > (5) Another point that I do not understand is why SINO’s inference time is dozens of times larger than that of the FNO. What makes SINO so expensive at inference? Why is its cost in almost the same ballpark as the numerical solver itself? In the earlier version of the paper, we could see that SINO was much closer to the numerical method than to the other data-driven surrogates in terms of inference time, which raises concerns about its practical usefulness.  If surrogate models are almost as slow as the numerical solvers, the motivation for using neural surrogates for PDEs largely disappears.

---

> > > ### Author Response · Authors · 2025-11-26
> > > **Response to Reviewer vLmH New Comment [1/3]**
> > >
> > > Thank you for your response.
> > >
> > > **Question 3: On the difficulty of the benchmarks.**
> > >
> > > **Reply:** Thank you for your follow-up comments. Based on your initial review (*"The authors should have evaluated SINO on more challenging benchmarks such as PDE-Gym or PDEBench, which include non-periodic or discontinuous dynamics."*) and your latest comments, we would like to clarify two concrete points:
> > >
> > > 1. **Viscosity in the NSE setup.**
> > > In PDEBench, the incompressible NSE benchmark uses a viscosity of $\nu = 0.01$, and PDE-Gym uses $\nu = 4\times 10^{-4}$. Nevertheless, our paper uses a much smaller viscosity of $\nu = 1\times 10^{-5}$, combined with non-smooth initial conditions and OOD generalization tests. This corresponds to a much higher Reynolds number and stricter long-horizon stability requirements, **making our NSE benchmark in our paper more challenging than the setups in PDEBench and PDE-Gym**.
> > >
> > >
> > > 2. **Non-periodic, nonlinear, time-dependent Burgers case.**
> > > **In PDE-Gym, all non-periodic benchmarks are linear or stationary, and in PDEBench, all benchmarks use periodic boundaries.** In contrast, we consider a Burgers problem that is **non-periodic, nonlinear, and time-dependent**, with mixed boundary conditions. This setup is chosen to address the concern about non-periodic dynamics, and it is more challenging than the benchmarks you mentioned.
> > >
> > > Beyond these specific comparisons, we would also like to **share our understanding of “challenging”.** According to well-known surveys on neural operators and AI4PDE [1, 2], the community has pointed out several open problems, including (but not limited to):
> > >
> > > 1. data scarcity,
> > > 2. unknown physics,
> > > 3. complex geometries,
> > > 4. out-of-distribution (OOD) generalization,
> > > 5. non-smooth fields,
> > > 6. high-Reynolds-number / high-speed turbulent flows,
> > >    …
> > >
> > > SINO is designed to address challenges (1), (2), (4), and (5) within a single framework:
> > >
> > > - **Data scarcity (1):** SINO is trained with only 2–5 trajectories, yet achieves SOTA performances.
> > > - **Unknown physics (2):** SINO does not require any explicit knowledge of the governing PDEs.
> > > - **OOD & non-smooth fields (4, 5):** In Section 5 (Fig. 6), SINO generalizes well to OOD initial conditions with significantly non-smooth fluid fields.
> > >
> > > **To our knowledge, no existing neural operator simultaneously tackles data scarcity, unknown physics, OOD generalization, and non-smooth fields like SINO.** While SINO does not resolve all open challenges, we believe that demonstrating strong performance under the simultaneous presence of challenges (1), (2), (4), and (5), and taking a meaningful step forward for the AI4PDE community.

---

> > > ### Author Response · Authors · 2025-11-26
> > > **Response to Reviewer vLmH New Comment [2/3]**
> > >
> > > **Question 4: Why do baselines fail while SINO succeeds? Concerns about a fair and fully specified training setup.**
> > >
> > > **Reply:** Thank you for your follow-up questions. Below, we explain **(i) why SINO works well in our regime**, **(ii) why data-driven neural operators fail**, and **(iii) whether using the same time-stepping and RK4 schemes can achieve comparable performance.**
> > >
> > > ---
> > >
> > > 1. **Why does SINO work in our setting?**
> > >
> > > SINO leverages a spectral-inspired architecture that yields a **compact yet expressive neural operator** with a favorable balance between approximation accuracy and generalization (Theory in Sec. 4). As shown in Fig. 4 (Sec. 5.2), the learned operators ensure a clear physical interpretation, which goes beyond other black-box operators.
> > >
> > > This is also **consistent with the strength you already highlighted**:
> > >
> > > > “By structuring the model around Fourier transforms and multiplicative nonlinear interactions, SINO incorporates strong physics-motivated priors (e.g., global coupling in the frequency domain, multiplicative nonlinearities akin to convection terms). These choices may help mitigate overfitting in low-data settings.”
> > >
> > > We believe this succinctly captures why SINO can outperform purely data-driven methods **specifically in the low-data regime** considered in our work.
> > >
> > > ---
> > >
> > > 2. **Why do FNO and other data-driven methods fail in our setting?**
> > >
> > > Firstly, we would like to reiterate that our work focuses on operator learning in a limited-data setting. The main reason is that **current data-driven neural operators are data-hungry**: they require lots of training trajectories (often more than 1k) to achieve good generalization. In our experiments, we **follow the standard FNO training pipeline** from the original FNO paper; **the only difference is the number of available training trajectories**, where only 2-5 trajectories are available.
> > >
> > > To verify that our comparison is fair and that our training pipeline works well, we train FNO with different numbers of trajectories on the same experiment (E2):
> > >
> > > **Table A: FNO performance with different numbers of training trajectories on E2**
> > >
> > > | # of trajectories | Relative $\ell_2$ error |
> > > |-------------------|-------------------------|
> > > | 5 (our setting)   | 0.5210 |
> > > | 50  | 0.1059 |
> > > | 200   | 0.0212 |
> > > | 1000 | 0.0112 |
> > >
> > >
> > > When trained with 1000 trajectories, the FNO results are **aligned with the original FNO paper**, confirming that our implementation and training pipeline are standard and correct. This shows that FNO’s poor performance is caused by data scarcity and overfitting, rather than by our experimental setup.
> > >
> > >
> > > ---
> > >
> > > 3. **Could an FNO with the same time stepping and RK4 scheme match SINO’s performance?**
> > >
> > > Firstly, we emphasize that the core advantage of SINO over FNO does **not** come from using RK4 time-stepping alone, but from its **strong physics-inspired inductive bias**, as discussed in Sec. 3.4 and Theorem 4.
> > >
> > > To further verify this, we trained FNO variants under the **same time-stepping configuration** as SINO (same $\Delta t$ and land RK4 scheme like SINO). The results on experiment E4 are summarized below:
> > >
> > > **Table B: Comparison of RK4-FNO and SINO on E4**
> > >
> > > | Model |   Test Error | Parameters | Training Time (min) | GPU Memory (MB) |
> > > |--| --:|--:|--:|--:|
> > > | FNO-channel-4|  0.3603| 10,021 | 58 | 1,758 |
> > > | FNO-channel-8 |  0.1905| 38,249 |60 | 1,792 |
> > > | FNO-channel-12|  0.0648|84,941 |61 |1,824 |
> > > | FNO-channel-36|  0.0923|752,837|63|2,068 |
> > > | **SINO** | **0.0031** | 9278 |61 |1,522|
> > >
> > > Even FNO is trained with the **same time stepping and RK4 scheme**, and its parameter count is increased by nearly an order of magnitude, **test error of FNO remains one order of magnitude worse** than SINO. This indicates that the performance gap cannot be explained by RK4 time-stepping; it is primarily due to SINO’s architecture and its physics-inspired inductive bias.

---

> > > ### Author Response · Authors · 2025-11-26
> > > **Response to Reviewer vLmH New Comment [3/3]**
> > >
> > > **Question 5：About inference efficiency.**
> > >
> > > **Reply:** Thank you for your comments.
> > >
> > > 1. **Why is SINO slower than FNO at inference?**
> > >
> > > The main reason is that SINO and FNO learn different objects. FNO is designed to directly learn the solution map from $t$ to $t + \Delta t$ (DOL). In contrast, SINO learns the differential operator on the right-hand side of the PDE and then integrates it with RK4 time-stepping and a smaller step size (POL). As a result, SINO needs more internal updates than FNO, leading to a higher inference cost, which is consistent with other works that follow a POL-style formulation.
> > >
> > > 2. **Does SINO violate the motivations of neural operators?**
> > >
> > > **In the regime with limited data and unknown physics:**
> > >
> > > - **Traditional numerical solvers** require an explicit knowledge of the PDEs. When the physics is unknown, classical solvers cannot work.
> > >
> > > - **Purely data-driven surrogates** such as FNO require lots of trajectories to generalize well. As we showed in Question 4 (Table A), they quickly overfit when only limited data are available.
> > >
> > > In this setting, SINO first learns a surrogate operator from only a few trajectories under unknown physics and serves as a teacher to train **SINO-FNO**, which achieves both **high accuracy and fast inference**. As a result, the overall SINO pipeline is accurate, fast, data-efficient, and physics-agnostic. **Therefore, SINO not only does not violate the motivations of neural operators, but actually expands the application scope of current methods.**
> > >
> > >
> > > [1] Azizzadenesheli, K., Kovachki, N., Li, Z., Liu-Schiaffini, M., Kossaifi, J., & Anandkumar, A. (2024). Neural operators for accelerating scientific simulations and design. Nature Reviews Physics, 6, 320–328. https://doi.org/10.1038/s42254-024-00712-5
> > >
> > > [2] Zhang, X., Wang, L., Helwig, J., Luo, Y., Fu, C., Xie, Y., Liu, M., Lin, Y., Xu, Z., Yan, K., Adams, K., Weiler, M., Li, X., Fu, T., Wang, Y., Strasser, A., Yu, H., … Ji, S. (2025). *Artificial Intelligence for Science in Quantum, Atomistic, and Continuum Systems*. Foundations and Trends® in Machine Learning, 18(4), 385–912.

---

> > > > ### Comment · Reviewer_vLmH · 2025-11-26
> > > >
> > > > I thank the authors for the additional clarifications. However, I believe that most of my key concerns remain insufficiently addressed.
> > > >
> > > > Could the authors please clarify the following passage:
> > > >
> > > > *The main reason is that SINO and FNO learn different objects. FNO is designed to directly learn the solution map from … to … (DOL). In contrast, SINO learns the differential operator on the right-hand side of the PDE and then integrates it with RK4 time-stepping and a smaller step size (POL).*
> > > >
> > > > In particular, I would appreciate clarification on the following points:
> > > >
> > > > - Is FNO trained and evaluated on the DOL discretization, while SINO is trained and evaluated on the POL discretization?
> > > >
> > > > - What precisely is meant by “learning the right-hand side of the PDE” in the context of SINO? On which time grid is the RHS learnt?
> > > >
> > > > I find it difficult to attribute the very large performance gap between SINO and all other models widely used in the community solely to inductive bias. I therefore ask the authors once again to provide an exact description of how FNO is trained and how SINO is trained, including the time step dt used at both training and inference for each model.
> > > >
> > > > -  The use of small-step time-stepping is not a novel technique. A forward pass with a small dt effectively corresponds to a local linearization of the dynamics over a short time interval. However, this fine time-stepping behavior must be learned from data, and I do not believe it can be reliably inferred solely from observations on the coarse DOL grid.
> > > >
> > > > - This is the source of my concerns. It is also possible that I did not fully understand the core idea behind your training and inference procedures.
> > > >
> > > > ____
> > > >
> > > > Other comments:
> > > >
> > > > - I strongly disagree with the claim that using a viscosity of 1e-5 makes this benchmark more difficult than, for example, those in PDEGym. According to the Appendix of PDEGym paper, the authors rely on the AZEBAN spectral hyperviscosity code, where viscosity is applied **to the higher modes**. In this setting, a direct comparison of difficulty is not straightforward. The authors could inspect the dynamics of several trajectories to assess the level of difficulty more carefully.
> > > >
> > > > - To train SINO, one requires data generated by a numerical solver, which in turn requires the underlying physics to be simulated. Thus, the method still fundamentally relies on access to the governing physics for data generation.

---

> ### Author Response · Authors · 2025-11-26
> **Please Stop Wasting Our Time with Your GPT**
>
> Dear Reviewer,
>
> Based on your original review and follow-up comments, we are highly confident that your words are **fully AI-generated** (https://iclr.pangram.com/). The follow-up questions generated via LLM are meaningless and lack factual grounding. In our view, your behavior is wasting our time and is also not aligned with the ICLR reviewer policy, and **papers on which you co-authored may be rejected**.
>
> We have reported your behavior to AC and SAC on Nov. 17. If you have any other questions, **please read our paper by yourself first**, and we will be happy to answer them.
>
> Best regards,
>
> The Authors

---

> > ### Comment · Reviewer_vLmH · 2025-11-26
> >
> > I am not sure what lead to your sudden decision. Honestly, LLMs are used for rephrasing only, just like all of your comments above. I am happy to discuss the AC about this case, as well as your paper and the results. I wish you all the best in the review process.

---

> ### Author Response · Authors · 2025-11-26
>
> 1. Given $u_t$, SINO predicts the right-hand side of the PDE $\mathcal{R}[u_t]$,
> which is then integrated with an RK4 scheme to obtain
> $u_{t+\Delta t}$ (the next time frame), like other classical methods. During training, SINO is learned on the POL grid. At inference time, SINO is rolled out with a small time step $\Delta t$ in line with traininng phase, while all models (POL and DOL) are evaluated on the DOL discretization for fair comparison, since the spatiotemporal field represented on the DOL grid is contained within that of the POL grid.
>
> 2. Hyperviscosity methods are used to enhance numerical stability while preserving large-scale features, and will not enhance the turbulence intensity.
>
> 3. We use a numerical solver solely to create a controlled benchmark where the ground-truth physics is known to the evaluator but unknown to the learning method, so that we can rigorously assess correctness and generalization, which is a standard protocol in the literature on symbolic regression and scientific discovery.

---

> > ### Comment · Reviewer_vLmH · 2025-11-26
> >
> > Leaving aside your accusations, I will continue the discussion.
> >
> > - Could you answer the second question about the FNO? Is the FNO trained on POL (just like the SINO) or on DOL? The main purpose of my comments was to understand whether the authors trained the models on POL or DOL grids. It is hard to extract this peace of information from the paper.

---

> ### Author Response · Authors · 2025-11-26
>
> In our paper, FNO is trained on DOL because FNO belongs to data-driven methods. We use DOL grid to align with its original settings. And we also added an additional result to train FNO on POL with RK4 in our response to you.
>
> > Firstly, we emphasize that the core advantage of SINO over FNO does **not** come from using RK4 time-stepping alone, but from its **strong physics-inspired inductive bias**, as discussed in Sec. 3.4 and Theorem 4. To further verify this, we trained FNO variants under the **same time-stepping configuration** as SINO (same $\Delta t$ and land RK4 scheme like SINO). The results on experiment E4 are summarized below:
>
> **Table B: Comparison of RK4-FNO and SINO on E4**
>
> | Model |   Test Error | Parameters | Training Time (min) | GPU Memory (MB) |
> |--| --:|--:|--:|--:|
> | FNO-channel-4|  0.3603| 10,021 | 58 | 1,758 |
> | FNO-channel-8 |  0.1905| 38,249 |60 | 1,792 |
> | FNO-channel-12|  0.0648|84,941 |61 |1,824 |
> | FNO-channel-36|  0.0923|752,837|63|2,068 |
> | **SINO** | **0.0031** | 9278 |61 |1,522|
>
> > Even FNO is trained with the **same time stepping and RK4 scheme**, and its parameter count is increased by nearly an order of magnitude, **test error of FNO remains one order of magnitude worse** than SINO. This indicates that the performance gap cannot be explained by RK4 time-stepping; it is primarily due to SINO’s architecture and its physics-inspired inductive bias.
>
> **PS:** According to several AI-detected tools, your reviews are detected as fully AI-generated rather than primarily written by a human and polished by AI. To my knowledge, the ICLR organizers also use the report provided by Pangram (https://iclr.pangram.com/) to judge AI-generated reviews. **If I have misjudged, I sincerely apologize.**

---

> > ### Comment · Reviewer_vLmH · 2025-11-26
> >
> > I believe there should be no fundamental difference between FNO and SINO in terms of how they are trained. I believe this is precisely why the relative error difference in the main table (for FNO and all other “DOL” models) is so large. I explicitly raised this already in my first comment:
> >
> > *"This means that SINO performs 100–1000× more effective temporal updates per trajectory, producing thousands of additional training samples and much higher temporal resolution"*.
> >
> > While this may have overstated the exact number of additional effective samples, the fact remains that one model is trained on the POL (fine) grid and the others on the DOL (coarse) grid. This raises serious concerns and, in my opinion, makes the comparisons to FNO and the other models fundamentally unfair.
> >
> > Given this issue, along with the other concerns I have previously mentioned, I will maintain my current score.
> >
> > I nevertheless wish you all the best in the remainder of the review process!
> >
> > ___
> >
> > **PS**: As I mentioned, LLMs are used for rephrasing my thoughts. A comment such as "Please Stop Wasting Our Time with Your GPT", is unprofessional, given the fact that I asked reasonable and very explicit questions.

---

> ### Author Response · Authors · 2025-11-26
>
> 1. We use FNO with the DOL time step to align with its original setting, which is a standard process for conducting baselines.
>
> 2. **The results of FNO trained with POL time step are provided in our response twice.**
>
> > Even FNO is trained with the **same time stepping and RK4 scheme**, and its parameter count is increased by nearly an order of magnitude, **test error of FNO remains one order of magnitude worse than SINO.**
>
> Therefore, we are really curious about why you still believe that *different time step is precisely why the relative error difference in the main table (for FNO and all other “DOL” models) is so large.*

---

### Official Review · Reviewer_vCqk · 2025-11-01

**Soundness:** 4
**Presentation:** 3
**Contribution:** 4
**Rating:** 8
**Confidence:** 3

**Summary:**

This paper proposes Spectral-Inspired Neural Operator (SINO), a neural PDE solver that learns complex physical dynamics from only 2–5 trajectories without requiring explicit PDE terms or residual supervision.
SINO combines several frequency-domain components: a Spectral Learning Block with Freq2Vec to learn frequency-wise multipliers, a Linear Operator Block to mix derivative-like channels, a Π-block for nonlinear multiplicative interactions with a 2/3 low-pass de-aliasing, and RK4 time-stepping for temporal evolution.
The model shows strong results on 2D and 3D PDE benchmarks, outperforming data-driven baselines trained on orders of magnitude more data, and demonstrating operator-level generalization to out-of-distribution (OOD) initial conditions.

**Strengths:**

The main strength of this paper lies in its data efficiency and physics-inspired inductive bias. By operating in the frequency domain and embedding derivative-like interactions through the pi block, the model implicitly encodes structural knowledge of PDE dynamics while remaining fully data-driven. This architectural design allows SINO to achieve remarkable performance using only a handful of training trajectories, outperforming other data-driven solvers that require hundreds or thousands of samples. Moreover, the model demonstrates impressive stability and accuracy over long time horizons, effectively suppressing compounding and high-frequency errors. The experiments show that SINO generalizes beyond the training distribution, accurately capturing phase and shape evolution of unseen initial conditions, suggesting a level of operator-level generalization. The methodology is clearly motivated and systematically described, and the experimental validation convincingly supports the paper’s central claims. Overall, the paper represents a meaningful advance in operator learning under data scarcity and physics-agnostic conditions.

**Weaknesses:**

Despite its strong empirical results, the paper has several limitations that should be addressed. First, the interpretability claims are primarily qualitative: while Figure 5 suggests that the Π-block features align with ground-truth differential operators, the paper provides no quantitative metrics (e.g., correlation or projection scores) to substantiate this claim. This weakens the argument that SINO learns physically meaningful spectral structures. Second, the 2/3 de-aliasing rule is fixed throughout the experiments, and the sensitivity of performance to the cutoff ratio is not explored. Understanding how different cutoff ratios affect accuracy and stability would clarify the robustness of the method across resolutions. Third, all theoretical and empirical results assume periodic boundaries and spectral grids, leaving it unclear whether the approach generalizes to non-periodic or irregular domains. Finally, while the ablation study compares RK4 and Euler integration, it does not analyze the effect of varying the RK4 step size itself, making it difficult to assess temporal stability more generally. These limitations do not invalidate the contribution but do restrict the generality and interpretability of the results.

**Questions:**

- If the RK4 integrator is retained, does a coarser time step maintain long-horizon stability? The paper shows that Euler integration reduces accuracy, but step-size sensitivity within RK4 is not reported.
- Are the pi-block and linear features consistently aligned with ground-truth operators across random seeds and resolutions, or is the alignment case-dependent?
- How sensitive is the model to the chosen 2/3 cutoff rule? Would alternative values (e.g., 0.5 or 0.8) change stability or accuracy significantly?
- Can the proposed method extend to non-periodic boundary conditions, or is the formulation inherently restricted to Fourier-periodic domains?

---

> ### Author Response · Authors · 2025-11-21
> **Response to Reviewer vCqk**
>
> Thank you for your time and valuable feedback, as well as the positive comments on our motivation, experiments, and presentation.
>
> **Weakness 1: Quantitative validation for the interpretability claims.**
>
> **Reply:** Great suggestion! To address this concern, we have incorporated correlation analysis as a quantitative metric to evaluate the interpretability of the results. Specifically, as visualized in **Figure 4** (**Section 5.2**) of the revised paper, the learned feature maps demonstrate strong alignment with the ground-truth operators, achieving high correlation coefficients of **0.96**, **0.85**, and **0.99** for $\mathbf{u}_x$, $\partial_x \omega$, and $\Delta \omega$, respectively. These results quantitatively confirm that our SINO model effectively learns the underlying physical meanings.
>
>
> **Weakness 2 & Question 3: Sensitivity analysis regarding the de-aliasing cutoff ratio.**
>
> **Reply:** Thank you for your comment! Following your suggestion, we conducted additional experiments using alternative cutoff ratios (0.5 and 0.8) on cases E2-E5 to analyze the sensitivity. The results are presented in **Table A** below and are also included as **Table 8** in the revised paper (**Appendix C.8**).
>
>
> **Table A: Relative $\ell_2$ error comparison with different de-aliasing cutoff ratios**
>
> | De-aliasing Cutoff | E2 | E3 | E4 | E5 |
> | :--- | :---: | :---: | :---: | :---: |
> | 2/3 (Original) | 0.0110 | 0.0303 | 0.0031 | 0.0171 |
> | 0.8 | 0.0123 | 0.0298 | 0.0036 | 0.0185 |
> | 0.5 | 0.0061 | 0.0474 | 0.0039 | 0.0261 |
>
> From the results in **Table A**, we observe that while the de-aliasing cutoff ratio has a marginal impact on the relative $\ell_2$ error, SINO maintains consistent stability and robustness across different settings. In this paper, we adopt the 2/3 rule to align with the standard hyperparameter choice in spectral methods.
>
>
> **Weakness 3 & Question 1: Analysis on varying the RK4 step size to assess temporal stability.**
>
> **Reply:** We performed a sensitivity analysis by varying the RK4 time step size ($\Delta t$) on the E3. The detailed results are summarized in **Table B** below and are also included as **Table 9** in the revised paper (**Appendix C.9**).
>
> **Table B: Performance (Relative $\ell_2$ Error) on E3 with varying RK4 step sizes**
>
> | Step Size ($\Delta t$) | Relative $\ell_2$ Error |
> | :---: | :---: |
> | 0.005s (Original) | 0.0303 |
> | 0.05s | 0.0223 |
> | 0.5s | NaN |
>
> From **Table B**, we observe that SINO maintains robust performance and long-horizon stability for time steps up to **0.05s**. However, as expected with explicit integrators (like RK4), an excessively large step size (e.g., **0.5s**) leads to NaN. This confirms that our method is stable within a reasonable range of coarse time steps.
>
>
>
> **Question 2: Robustness of learned operators across different settings.**
>
> **Reply:** Thank you for your suggestions! To address this, we have visualized the learned feature maps corresponding to $\Delta \omega$ across datasets E2–E5 in **Figure 11** of the revised paper (**Appendix C.7**). The results show that the learned features consistently align closely with the ground-truth operators in all cases, providing strong evidence for the **robustness** of our method across different settings.
>
> **Question 4: Applicability to non-periodic boundary conditions.**
>
> **Reply:** Insightful question! To address this, we have introduced a more challenging dataset (**E8**) involving the 2D Burgers' equation with **mixed boundary conditions** (Dirichlet BCs on top/bottom and Periodic BCs on left/right). Extending SINO to non-periodic boundaries is natural: we simply replace the FFT with an appropriate spectral transform (e.g., Discrete Sine Transform (DST)) to handle the non-periodic components. The detailed settings and results are included in the revised paper (**Table 1, Section 5.2**).
>
> **Table C: Relative $\ell_2$ error on E8 (MixedBC)**
>
> | Model | Relative $\ell_2$ Error (E8, MixedBC) |
> | :--- | :---: |
> | PeRCNN | 0.0170 |
> | PeSANet | 0.0967 |
> | CROP | 0.9833 |
> | CNext | 0.2007 |
> | FactFormer | 0.5908 |
> | FFNO | 0.2837 |
> | FNO | 0.4495 |
> | SNO | 1.0001 |
> | LSM | 0.0962 |
> | PDENet 2.0 | 0.2210 |
> | P$^2$C$^2$Net | 0.0138 |
> | **SINO (ours)** | **0.0002** |
>
> As shown in **Table C**, SINO achieves a relative $\ell_2$ error of **0.0002**, significantly outperforming the baseline methods. This demonstrates SINO's flexibility and superior performance in handling non-periodic boundary conditions.

---

### Author Response · Authors · 2025-11-17
**Flagging Reviewer vLmH’s Review as Potentially Fully AI-Generated**

Dear **Area Chair**,

In response to the ICLR official statement, we decide to report the behavior of reviewer `vLmH`. According to the LLM reports on [https://iclr.pangram.com/](https://iclr.pangram.com/) and our careful examination, we are very confident that `vLmH` submitted a ***fully AI-generated*** review with a score of 2 and confidence 4. We believe this behavior is inappropriate and violates the ICLR reviewer policy.

We would like to inquire whether the authors can disregard this fully AI-generated review. If not, we are also willing to address some of the valuable points raised.


Sincerely,

The Authors

>**From ICLR2026 official Twitter:**
We are aware of low-quality and LLM-generated reviews and are currently deliberating on appropriate courses of action. For now, authors who receive very poor quality or LLM-generated reviews should flag them to their ACs. We appreciate the community's efforts in reporting these!

Here we list some evidence below:

1. Both Pangram and GPTZero classify the review of `vLmH` as **fully AI-generated with high confidence, rather than as human-written text and polished by LLMs.**

2. **Strange Request.** `vLmH` asks us to report the information about “data augmentation”. However, in operator learning, data augmentation is rarely used, and none of the main baselines we compare against use such strategies. A reviewer with expertise in this area should know this, suggesting that this comment is generated via LLM.

3. **Inconsistency between the original review and the subsequent discussion.**

**3.a.** In the original review, `vLmH` clearly wrote that *SINO is trained and integrated with a fine time step (Δt ≈ 1e−3–1e−2) and Runge–Kutta 4 (RK4) integration, while the data-driven baselines (FNO, FFNO, CNext, etc.) are trained with much coarser Δt (0.2–1) and single-step rollouts.* However, in the follow-up comments, `vLmH` asked about the choice of Δt and stated that *It is hard to extract this peace of information from the paper.* This clear inconsistency between the original review and the subsequent discussion makes us even more convinced that **the original review was fully AI-generated** and **the reviewer himself does not fully understand what the LLM wrote**, whereas some of the later replies were written manually after we pointed out his LLM-generated content.

**3.b.** `vLmH` first stated: *I apologize for my earlier statement about having an N-times larger number of effective samples*, while also claiming that *have overstated the exact number of additional effective samples* later. We believe that **his GPT-generated responses at different times contradict each other, possibly because of poor memory of GPT or opening of new chat sessions.**

---

### Comment · Reviewer_vLmH · 2025-11-26

Dear PCs and ACs,

The authors of the paper have accused me of generating all of my comments using an LLM. In my case, LLMs are used **only to rephrase** my own thoughts.

I believe the authors intentionally made some important details very difficult to understand. Their results suggest that only their approach works on the tested benchmarks, even though they compare against many different SOTA SciML models. It is, in my view, very suspicious that a relatively small change in inductive bias and time-stepping would make their model perform so much better than any other method, if everything was tested fairly.

For these reasons, I repeatedly asked the authors to provide the exact values of their parameters and to clearly explain what was actually done in the paper (as this is impossible to deduce from the manuscript alone). Already in my first review, I requested that they release all the code. Instead, they only provided partial codes. I additionally suggested which benchmarks should be tested and explained why the benchmarks they used are not particularly challenging. They addressed almost none of these concerns and gave very vague answers. In their most recent comments (after making the accusation), they described, in my view, a comparison setup between their model and the baselines that was systematically unfair. To further strengthen my concerns about this work, I note that in their first response to the reviews the authors stated that *“All methods, including SINO, FNO, FFNO, CNext, etc., are trained on exactly the same set of spatio-temporal snapshots”*. However, their later clarifications and the partially released code indicate that SINO is effectively trained and evaluated on a finer temporal grid.

These issues, together with the other concerns (i.e. **weak benchmarks and overall contribution**) that I have raised in my comments, is the reason for my low overall score.

I was accused of cheating, which I consider completely unfair and unprofessional. They even used the phrase "Please Stop Wasting Our Time with Your GPT". I have invested a significant amount of time in reviewing this paper, and I stand by my assessment. I am happy to defend my position and continue the discussion with a responsible AC.

---

> ### Author Response · Authors · 2025-11-27
>
> Given training data with time steps from $\{t_0,t_1,...,t_{2000}\}$, SINO uses fine time steps to construct loss (e.g., $t_1$ and $t_2$) while data-driven methods use the larger one (e.g., $t_\{10}$ and $t_\{110}$). However, data-driven methods can also use $t_\{11}$ and $t_\{111}$, $t_\{12}$ and $t_\{112}$, and others to construct the loss. Therefore, we claim that all methods, including SINO, FNO, FFNO, CNext, etc., are trained on exactly the same set of spatio-temporal snapshots. Could you please kindly let us know if you still have other questions?
>
> Furthermore, why do you ignore the results of FNO trained with POL time step (provided in our response), which clearly demonstrate that:
> > Even FNO is trained with the same time stepping and RK4 scheme, and its parameter count is increased by nearly an order of magnitude, the test error of FNO remains one order of magnitude worse than SINO.
>
> ---
>
> Regarding the statement *“I believe the authors **intentionally** made some important details very difficult to understand”*, we would like to clarify that **this is a purely subjective and unprofessional accusation**. **We have already uploaded our code, and if needed, we are happy to assist you in running SINO and other baselines to ensure reproducibility.**
>
> **In contrast, your fully-AI generated reviews are vague and inconsistent.** Even if you claim that you participated in the reviewing process, it appears that the prompts you gave the LLM were incomplete or vague, resulting in the confusing reviews. Moreover, it seems that you did not check or edit the generated content, which is why it was flagged as fully AI-generated.
>
> ---
>
>
> Moreover, you said  *It is, in my view, very suspicious that a relatively small change in inductive bias and time-stepping would make their model perform so much better than any other method, if everything was tested fairly.*
>
> **Our reply**: You are dismissing a promising work. **If you do not trust our results, you are welcome to run the code we provided.** **You may also test the performance of FNO**, and **we are confident its accuracy is at least an order of magnitude worse than ours.** If needed, **we can also provide you with the training code for FNO**, including both DOL and POL+RK4 implementations. Furthermore, **we strongly disagree with your comment that SINO only has “a relatively small change in inductive bias.”** Our model architecture significantly differs from all existing neural operators. This architectural novelty (rather than time-stepping, we never and ever regarded the small dt + RK4 as our novelty) is the foundation of SINO for achieving interpretability and strong OOD generation ability.

---

### Author Response · Authors · 2025-11-29
**Response to AC, SAC and PC**

Dear ACs,

Thank you for considering our paper. We provide a summary of our responses.
### **Contribution**
**Background:** Existing neural operators require either **large training data** (e.g., FNO) or **rich physics-based priors** (e.g., PINO).

**Our setting:** Training neural operators from **limited data** (e.g., 2–5 trajectories) and **unknown physics**.

**Highlighted results:**
1. SOTA performance, with improvements of **1–2 orders of magnitude in accuracy**. SINO is the first physics-aware method capable of simulating globally coupled systems without requiring explicit PDE terms.
2. With only **5** training trajectories, SINO outperforms data-driven methods trained on **1000** trajectories and retains strong **out-of-distribution** generalization ability.
### **Major Concerns**
**1. Experiments**

**a.** **Parameter count:** Reviewer `ZG9x` comments that *the parameter count of baselines is significantly larger than SINO.*

**Reply:** SINO and other baselines have fundamentally different capacity–complexity trade-offs, so **strict parameter matching would depart significantly from the original papers and be unfair to those baselines**. We also provide parameter-matched variants, and they perform far worse than the results reported in our paper.

**b.** **Time-step:** The reviewer `vLmH` believes that SINO can use additional training samples because of its fine step size.

**Reply:** **1) All methods use the same number of temporal frames**, ensuring equal data utilization (verified by the updated code). For instance, SINO uses fine time steps to construct loss (e.g., $t_1$ and $t_2$) while data-driven methods use the larger ones (e.g., $t_\{10}$ and $t_\{110}$). However, data-driven methods can also use $t_\{11}$ and $t_\{111}$, $t_\{12}$ and $t_\{112}$. **2) Our setting follows the original configurations of each baseline** to ensure a fair comparison. **3) We also trained FNO with RK4 and aligned $\Delta t$**, where FNO remains significantly weaker than SINO.

**c.** **Hyperparameters search:** Our study involved a **comprehensive hyperparameter exploration**. **For each method–data pair**, we tested **tens of** distinct key hyperparameter configurations. In total, we trained **over 1,000 neural operators.** The training loss curves clearly show that each baseline was trained until convergence.

**d. Compared to physics-based methods:** Physics discovery methods are often confined to systems governed by local operators and struggle with globally coupled systems without additional priors. We tested three discovery methods, including **PeRCNN**, **PESANet**, and the newly added **PDENet2.0**. Moreover, **Hybrid methods** rely on known physics. We added a new baseline **P$^2$C$^2$Net**, which **needs full knowledge of the governing equations**.

**e. Why SINO works:** SINO embeds a strong spectral inductive bias inspired by PDE structure, supported by theory and ablations. We also **released the full training code for SINO and FNO**. **We are confident that SINO consistently outperforms FNO by at least an order of magnitude**, and we **welcome the community to verify this, no matter whether this paper is accepted or rejected.**

**2. Novelty**

SINO is **fundamentally different** from existing operator-learning methods, as detailed in **Sec. 3.4**. Its strong few-shot performance and high OOD generalization **can not be achieved with an incremental modification**.

**3. Challenging Task**

Regarding the challenge of the task, we would also like to share our understanding of “challenging”. According to well-known surveys on neural operators and AI4PDE, **major open challenges** include:

1) Data scarcity
2) Unknown physics
3) Complex geometries
4) OOD generalization
5) High-Re or high-speed turbulence
6) Long-term rollout
 …

SINO successfully addresses challenges 1), 2), 4), and 6) within a single framework. Of course, we acknowledge that **a conference paper cannot resolve all open problems**.

**4. Additional results**

During rebuttal, we added four new baselines, a non-periodic case, and additional analyses.
### **Clarification**
We would like to report a serious concern regarding Reviewer `vLmH`. The reviews of `vLmH` appear to be **fully AI-generated** (as indicated by Pangram and GPTZero) and contain **confused requirements**, such as requesting data augmentation within hyperparameter search. During the discussion, despite our efforts to provide new results and clarifications, the reviewer `vLmH` **ignored our responses and continued to use LLMs to generate comments**, which even **contradicted with `vLmH`'s original review**. Moreover, we strongly disagree with Reviewer `vLmH`’s subjective allegation that we intentionally withheld experimental details. **In response, we have released the code for both SINO and FNO, and we welcome `vLmH` and everyone to run our codes; we are happy to provide assistance regardless of whether the paper is accepted.**

Best,

The Authors

---

### Meta-Review · Area_Chair_qUyc · 2026-01-08

**Summary:**

This paper presents a spectral-inspired neural operator for learning PDE dynamics from limited data (2-5 trajectories) without explicit knowledge of governing equations. The method introduces Freq2Vec for learning frequency-domain derivative multipliers. While the paper addresses an important problem and shows promising empirical results, the experimental setup raises fairness concerns that were not fully resolved. The use of RK4 time-stepping with small Δt provides SINO with structural advantages that are not adequately disentangled from the architectural contributions. Furthermore, the inference efficiency limitation is substantial for practical deployment. Given these weaknesses, the AC recommend rejection.

**Reviewer Concerns:**

Addressed concerns. For parameter count fairness, the authors provided ablation studies showing parameter-matched baselines still underperform. For quantitative interpretability, correlation analysis between learned features and ground-truth operators was added.

Outstanding concerns. The concern that SINO's POL formulation with RK4 integration provides inherent advantages over single-step baselines remains. SINO's computational overhead during inference is a significant practical limitation.

**Reviewer Scores:**

Reviewer ZG9x is concerned about the inference efficiency limitation and the questions about practical applicability persist. This reviewer did not engage in discussion, but based on the outstanding concerns, a significant score increase seems unlikely.

---

### Decision · Program_Chairs · 2026-01-26

Reject